# Privileged Information Distillation for Language Models

Emiliano Penaloza [1 2 3]  Dheeraj Vattikonda [1 2 4]  Nicolas Gontier [1]  Alexandre Lacoste [1]  Laurent Charlin [2 5]
Massimo Caccia [1]

## Abstract

Training-time privileged information (PI) can enable language models to succeed on tasks they would otherwise fail, making it a powerful tool for Reinforcement Learning (RL) in hard, long-horizon settings. However, transferring capabilities learned with PI to policies that must act without it at inference time remains a fundamental challenge. We study this problem in the context of distilling frontier models for multi-turn agentic environments, which typically hide their internal reasoning and expose only action trajectories. This breaks standard distillation pipelines, since successful behavior is observable, but the reasoning process is not. For this, we introduce $\pi$-Distill, a joint teacher-student objective that trains a PI-conditioned teacher and an unconditioned student simultaneously using the same model. Additionally, we also introduce On-Policy Self-Distillation (OPSD), an alternative approach that trains using RL with a reverse KL-penalty between the student and the PI-conditioned teacher. We show that both of these algorithms effectively distill frontier agents using action-only PI. Specifically, we find that $\pi$-Distill and, in some cases, OPSD, outperform industry standard practices (Supervised finetuning followed by RL) that assume access to full Chain-of-Thought supervision across multiple agentic benchmarks, models, and forms of PI. An arXiv version[1] of this manuscript is available online, and we also provide a beginner-friendly blog post.

[1]ServiceNow AI Research, Montréal, Canada [2]Mila – Quebec AI Institute, Montréal, Canada [3]Université de Montréal, Montréal, Canada [4]McGill University, Montréal, Canada [5]HEC Montréal, Montréal, Canada. Correspondence to: Emiliano Penaloza <emiliano.penaloza@mila.quebec>, Dheeraj Vattikonda <dheeraj.vattikonda@mail.mcgill.ca>.

*Proceedings of the $43^{rd}$ International Conference on Machine Learning*, Seoul, South Korea. PMLR 306, 2026. Copyright 2026 by the author(s).

[1]arxiv.org/abs/2602.04942

## 1. Introduction

Language Models (LMs) have the unique ability to converse, which provides a superior user interface and more straightforward interactions than other machine learning systems. Crafting good prompts, or equivalently, conditioning on the right context, remains essential to obtain the best possible performance from an LM. This raises the question of whether LMs can learn from informative prompts to generalize to less informative prompts. In other words, how can an LM learn to transfer training-time privileged information (PI) (Vapnik & Vashist, 2009; Lopez-Paz et al., 2016) to test tasks (that do not contain PI)?

Training-time PI can be particularly useful for Reinforcement Learning (RL) with LMs, where learning is contingent on the model's ability to first succeed on a task (Yue et al., 2025). That is, the model can leverage train-time PI to succeed at tasks it would otherwise fail, effectively enabling it to bootstrap its learning from these successful experiences. As noted above, the underlying challenge is transfer: training a model with PI and obtaining a test-time policy that retains those enhanced capabilities without PI.

In this work, we show that leveraging train-time PI is highly effective under certain conditions. For instance, conditioning a policy on PI can drastically shift the sampling distribution away from the unconditioned one, making transfer significantly harder. Additionally, we find that non-frontier models often struggle to accurately leverage PI and must explicitly learn to use it. To enable effective training in this setting, we introduce two complementary distillation objectives: Privileged Information Distillation ($\pi$-Distill), our main method, and On-Policy Self-Distillation (OPSD), an on-policy alternative. $\pi$-Distill adapts the typical teacher-student setup by using a single shared-parameter model in which the teacher has access to PI. Importantly, $\pi$-Distill trains the teacher and student jointly, enabling the teacher to learn how to use PI while actively mitigating distribution shift during transfer. OPSD similarly allows the teacher and student to share parameters, regularizing on-policy RL with a reverse KL penalty between the student and a PI-conditioned teacher.

We ground our work in the task of distilling frontier models for complex multi-turn agentic settings. Typically, the indus-

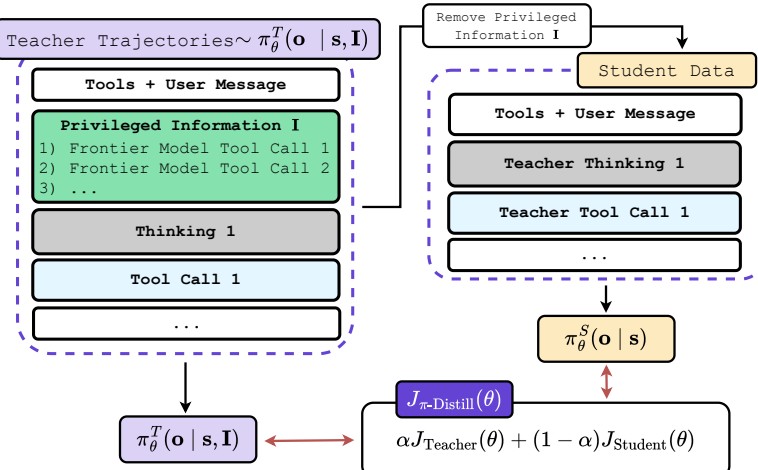

*Figure 1.* **Overview of the $\pi$-Distill framework.** (1) Successful trajectories (not shown) are collected from a frontier agent that exposes only actions while hiding its Chain-of-Thought. (2) These trajectories are transformed into training-time privileged information (PI) and used to sample using a PI-conditioned teacher policy $\pi_\theta^T(\mathbf{o} \mid s, \mathbf{I})$. (3) The PI-conditioned teacher and an unconditioned student $\pi_\theta^S(\mathbf{o}|s)$ share parameters and are trained jointly, enabling transfer of privileged knowledge to a test-time policy that acts without PI.

*Figure 2.* **Results for `Qwen3-8B` on TravelPlanner and $\tau$-Bench retail.** The dashed line separating SFT w/ CoT + RL denotes that this method is not a required baseline, as all other methods do not rely on frontier-model CoT traces. We find that *both* $\pi$-Distill and OPSD substantially outperform all baselines in this setting.

try standard for these tasks involves Supervised Fine-Tuning (SFT) on frontier model outputs followed by Reinforcement Learning (RL). Unfortunately, some model providers restrict important information, most notably the model's full Chain-of-Thought (CoT) reasoning traces (OpenAI et al., 2024), providing only a summary alongside the action they intend to take. This opacity undermines standard distillation methods, as we can observe what successful agents do but not how they reason about it.

We find that $\pi$-Distill is highly effective at mitigating the lack of CoT, outperforming industry standards that assume full CoT access, with this holding for OPSD in some cases. We demonstrate this on two agentic tool-use environments, Travel Planner (Xie et al., 2024) and $\tau$-Bench retail (Yao et al., 2024), while showing proficient Out-of-Domain generalization on the 7 tool-use environments provided by GEM (Liu et al., 2025b) and $\tau$-Bench airline. Our findings are consistent across three models from two distinct families.

Finally, we transform frontier model trajectories into three varying types of PI, each with different information density, providing varying amounts of utility and inducing different degrees of distributional shift between student and teacher. We use this variation to analyze the critical factors for training with PI, finding that for $\pi$-Distill, maximizing PI utility while simultaneously mitigating the student-teacher distributional gap and preventing collapse is essential for effective learning. While information content is more important for OPSD.

## 2. Background

**Agentic Interaction as an MDP.** We formalize long-horizon, multi-turn agentic environments as a Markov Decision Process (MDP). In this setting, a policy $\pi_\theta(\cdot \mid \mathbf{s})$ interacts with an environment over extended sequences of actions. To simplify notation, we let $\mathbf{s}$ represent the *evolving interaction context*, which aggregates all information available to the model at a given point: the initial user prompt, the model's past outputs, and all environment responses. As the agent acts, the context updates via a transition function $\mathbf{s}_{t+1} \sim P(\cdot \mid \mathbf{s}_t, \mathbf{o}_t)$, where the environment appends its response to the current sequence to form the next state. As $\mathbf{s}$ encapsulates the interaction history, we use the following to simplify notation:

$$\pi_\theta(\mathbf{o} \mid \mathbf{s}) = \prod_{i=0}^{T} \pi_\theta(\mathbf{z}_i, \mathbf{a}_i \mid \mathbf{s}_{<i})$$

where $\mathbf{o} = (\mathbf{z}, \mathbf{a})$ consists of reasoning tokens $\mathbf{z}$ and action tokens $\mathbf{a}$.

**Reinforcement learning.** RL seeks to maximize the expected return obtained through interaction with an environment (Sutton et al., 1999):

$$J(\pi_\theta) = \mathbb{E}_{\substack{\mathbf{o} \sim \pi_\theta(\cdot \mid \mathbf{s}) \\ \mathbf{s} \sim P}} \Big[ R(\mathbf{o}, \mathbf{s}) \Big].$$

Here, $R(\mathbf{o}, \mathbf{s}) \in [-1, 1]$ denotes the reward assigned by the environment to the generated trajectory.

**Policy Optimization.** We optimize $J(\pi_\theta)$ using Group Relative Policy Optimization (GRPO) (Shao et al., 2024;

DeepSeek-AI et al., 2025), with the adjustments recommended by Yu et al. (2025) and Liu et al. (2025a). For each state $\mathbf{s}_i$, we sample a group of $G$ trajectories $\{\mathbf{o}_g\}_{g=1}^{G}$ according to the current sampling policy $\mu$ and the transition function $P$. For each token $k$ in trajectory $g$, we define the token-level importance ratio:

$$\rho_{g,k}(\theta) = \frac{\pi_\theta(o_{g,k} \mid \mathbf{s}_i, \mathbf{o}_{g,<k})}{\mu(o_{g,k} \mid \mathbf{s}_i, \mathbf{o}_{g,<k})}.$$

We define a group-relative advantage $A_{s,g}$ by comparing the return of trajectory $g$ to the average return of the sampled group, and use it to scale clipped importance-weighted policy updates. The GRPO objective then is:

$$J_{\text{GRPO}}(\theta) = \mathbb{E}_{\substack{\mathbf{o} \sim \pi_\theta(\cdot|\mathbf{s}) \\ \mathbf{s} \sim P}}\left[ \frac{1}{\sum_g K_g} \sum_{g,k} \min\left( \rho_{g,k} A_{\mathbf{s},g}, \text{clip}(\rho_{g,k}, 1-\epsilon, 1+\epsilon) A_{\mathbf{s},g} \right) \right]$$

where $K_g$ is the length of trajectory $g$ in tokens and $\epsilon$ is the clipping parameter. While the original GRPO objective typically employs a KL-penalty with respect to the base model, we drop this term as recent work shows it can hinder performance (Shah et al., 2026).

# 3. Methods

In this section, we introduce and motivate two algorithms for leveraging PI during training. Both methods use a teacher–student framework inspired by traditional distillation. However, the teacher and student share parameters, and only the teacher is conditioned on PI.

Privileged Information Distillation ($\pi$-Distill) learns from teacher-generated traces by jointly improving both the teacher and the student. In contrast, on-policy self-distillation (OPSD) [†] samples trajectories from the student policy and uses a reverse KL divergence between the student and the PI-conditioned teacher as a training penalty. Both algorithms are derived using a variational perspective, which we outline in detail in App. E and App. F.

## 3.1. Privileged Information Distillation ($\pi$-Distill)

**Motivation** A straightforward way to use PI for distillation is to condition a policy on PI to generate successful trajectories, which are then used for fine-tuning. However, this approach has two key limitations. First, base models do not automatically know how to exploit PI (see Section 7), needing to learn how to use it, before providing benefit. Second, even once the policy can exploit PI, we still need to transfer this behavior to a policy that must act without PI at test time.

---

[†] We adopt the naming "self-distillation" following concurrent work (Shenfeld et al., 2026; Zhao et al., 2026; Hübotter et al., 2026) that propose the same objective.

A naive solution is to first train a PI-conditioned policy and then distill its behavior into an unconditioned one. In practice, this sequential pipeline introduces several issues. It is unclear which checkpoint of the conditioned policy should be distilled, learning from its trajectories is off-policy and can be unstable, and training the two policies separately is computationally inefficient. Our early experiments confirm that this setup leads to suboptimal performance (see Figure 11).

To address these challenges, we propose Privileged Information Distillation ($\pi$-DISTILL), which trains both policies jointly within a single parameter-shared model. This allows the model to learn to exploit PI while simultaneously learning to act without it.

**Algorithm.** Our approach uses a single model with shared parameters $\theta$ that acts as both a teacher $\pi_\theta^{\text{T}}(\mathbf{o} \mid s, \mathbf{I})$ (conditioned on PI $\mathbf{I}$) and a student $\pi_\theta^{\text{S}}(\mathbf{o}|s)$ (operating without PI). We train both simultaneously using two objectives. The teacher objective trains the conditioned policy to maximize reward while maintaining proximity to the student policy:

$$J_{\text{Teacher}}(\theta) = \mathbb{E}_{\substack{\mathbf{o} \sim \pi_\theta^{\text{T}}(\mathbf{o}|s,\mathbf{I}) \\ \mathbf{s} \sim P}}\left[ R(\mathbf{o}, \mathbf{s}) \right] - \beta D_{\text{KL}}\left( \pi_\theta^{\text{T}}(\mathbf{o} \mid s, \mathbf{I}) \, \| \, \text{sg}\big(\pi_\theta^{\text{S}}(\mathbf{o}|s)\big) \right).$$

This objective samples trajectories from the teacher policy and updates it to increase reward while subject to a reverse KL penalty $D_{\text{KL}}\left( \pi_\theta^{\text{T}} \, \| \, \pi_\theta^{\text{S}} \right)$ controlled by $\beta$, where $\text{sg}(.)$ indicates the stop gradient operator. This objective serves two purposes: (i) it encourages the teacher to fit high-reward modes familiar to the student, making learning from its traces easier, and (ii) shared parameters promote transfer of the teacher's knowledge to the student, even without directly training the student.

The student objective trains the unconditioned policy to learn from the teacher's trajectories:

$$J_{\text{Student}}(\theta) = \mathbb{E}_{\substack{\mathbf{o} \sim \pi_\theta^{\text{T}}(\mathbf{o}|s,\mathbf{I}) \\ \mathbf{s} \sim P}}\left[ \frac{\pi_\theta^{\text{S}}(\mathbf{o}|s)}{\text{sg}\big(\pi_\theta^{\text{T}}(\mathbf{o} \mid s, \mathbf{I})\big)} R(\mathbf{o}, \mathbf{s}) \right] - \beta D_{\text{KL}}\left( \text{sg}(\pi_\theta^{\text{T}}(\mathbf{o} \mid s, \mathbf{I})) \, \| \, \pi_\theta^{\text{S}}(\mathbf{o}|s) \right).$$

This objective samples trajectories from the teacher (which has access to PI) but updates the student policy (which does not). This teaches the student to replicate the teacher's high-reward behavior without needing PI.

Combining the student and teacher terms gives us our final objective:

$$J_{\pi\text{-Distill}}(\theta) = \alpha J_{\text{Teacher}}(\theta) + (1 - \alpha) J_{\text{Student}}(\theta).$$

where $\alpha \in [0, 1]$ controls the balance between student and teacher focused learning.

*Table 1.* **Evaluation results on Travel Planner, $\tau$-Bench (Retail), and $\tau$-Bench (Airline)**. Shaded rows denote our methods. **Bold** values indicate the best performance within each model category, while underlined values indicate the second-best. Results show mean $\pm$ standard deviation across three random seeds. We find that both $\pi$-Distill and OPSD effectively leverage PI, consistently outperforming all baselines that lack access to frontier reasoning traces. Furthermore, both methods can surpass SFT w/ CoT + RL on TravelPlanner, with $\pi$-Distill also achieving superior performance on $\tau$-Bench.

| | Travel Planner | $\tau$-Bench Retail | $\tau$-Bench Airline (OOD) |
|---|---|---|---|
| *DeepSeek V3.1 Chat-671B* | | | |
| Base | 45.0% $\pm$ 3.78 | 51.3% $\pm$ 0.212 | 40.0% $\pm$ 0.161 |
| *R1-Distill-Llama-8B* | | | |
| Base | 0.00% $\pm$ 0.00 | 0.00% $\pm$ 0.00 | 0.00% $\pm$ 0.00 |
| SFT w/ CoT | 6.35% $\pm$ 1.27 | 15.2% $\pm$ 0.44 | 8.00% $\pm$ 5.29 |
| SFT w/o CoT | 2.40% $\pm$ 0.86 | 0.680% $\pm$ 0.17 | 0.670% $\pm$ 1.15 |
| SFT w/ CoT + RL | 12.4% $\pm$ 1.56 | 16.3% $\pm$ 1.49 | 7.33% $\pm$ 4.16 |
| SFT w/ CoT + On-Policy Self Distillation | 13.1% $\pm$ 1.09 | 14.5% $\pm$ 0.00 | 2.00% $\pm$ 0.00 |
| SFT w/ CoT + $\pi$-Distill $\pi^S$ ($\alpha = 0$) | 7.86% $\pm$ 1.75 | **18.6%** $\pm$ 0.50 | **10.0%** $\pm$ 2.00 |
| SFT w/ CoT + $\pi$-Distill $\pi^S + \pi^T$ ($\alpha = 0.5$) | 14.0% $\pm$ 1.63 | 18.3% $\pm$ 0.77 | 9.33% $\pm$ 3.06 |
| SFT w/ CoT + $\pi$-Distill $\pi^T$ ($\alpha = 1$) | **14.1%** $\pm$ 3.27 | 17.7% $\pm$ 0.77 | 7.33% $\pm$ 4.16 |
| *QWEN3-4B* | | | |
| Base | 17.6% $\pm$ 2.16 | 5.03% $\pm$ 1.88 | 2.21% $\pm$ 1.99 |
| SFT w/ CoT | 21.1% $\pm$ 1.94 | 12.4% $\pm$ 0.60 | 2.67% $\pm$ 1.15 |
| SFT w/o CoT | 20.8% $\pm$ 1.82 | 15.2% $\pm$ 1.70 | 4.00% $\pm$ 5.29 |
| RL | 25.1% $\pm$ 3.75 | 15.2% $\pm$ 0.17 | 5.33% $\pm$ 3.06 |
| SFT w/o CoT + RL | 23.3% $\pm$ 1.33 | 17.6% $\pm$ 2.69 | 5.33% $\pm$ 2.31 |
| On-Policy Self Distillation | 29.8% $\pm$ 1.14 | 23.1% $\pm$ 0.04 | 10.6% $\pm$ 6.57 |
| $\pi$-Distill $\pi^S$ ($\alpha = 0$) | 28.5% $\pm$ 4.35 | **25.3%** $\pm$ 0.60 | 8.00% $\pm$ 5.29 |
| $\pi$-Distill $\pi^S + \pi^T$ ($\alpha = 0.5$) | **33.8%** $\pm$ 6.85 | 22.6% $\pm$ 0.93 | 6.00% $\pm$ 2.00 |
| $\pi$-Distill $\pi^T$ ($\alpha = 1$) | 28.2% $\pm$ 6.27 | 22.5% $\pm$ 0.93 | **12.0%** $\pm$ 5.29 |
| SFT w/ CoT + RL | 26.4% $\pm$ 1.16 | 23.3% $\pm$ 3.02 | 6.67% $\pm$ 5.77 |
| *QWEN3-8B* | | | |
| Base | 23.6% $\pm$ 2.23 | 3.35% $\pm$ 1.47 | 6.40% $\pm$ 3.02 |
| SFT w/ CoT | 26.0% $\pm$ 2.27 | 16.5% $\pm$ 4.66 | 5.33% $\pm$ 1.15 |
| SFT w/o CoT | 29.8% $\pm$ 1.71 | 12.8% $\pm$ 0.77 | 6.00% $\pm$ 4.00 |
| RL | 27.5% $\pm$ 0.95 | 23.9% $\pm$ 0.44 | 6.67% $\pm$ 3.06 |
| SFT w/o CoT + RL | 31.3% $\pm$ 2.79 | 23.5% $\pm$ 4.27 | 6.00% $\pm$ 2.00 |
| On-Policy Self Distillation | 37.5% $\pm$ 1.53 | 27.3% $\pm$ 0.33 | **14.0%** $\pm$ 5.66 |
| $\pi$-Distill $\pi^S$ ($\alpha = 0$) | 40.7% $\pm$ 1.14 | **31.1%** $\pm$ 0.73 | 12.0% $\pm$ 6.00 |
| $\pi$-Distill $\pi^S + \pi^T$ ($\alpha = 0.5$) | 41.1% $\pm$ 7.24 | 30.6% $\pm$ 0.67 | 7.33% $\pm$ 1.15 |
| $\pi$-Distill $\pi^T$ ($\alpha = 1$) | **44.1%** $\pm$ 2.16 | 29.7% $\pm$ 0.33 | 9.33% $\pm$ 3.06 |
| SFT w/ CoT + RL | 32.3% $\pm$ 1.10 | 29.1% $\pm$ 2.14 | 8.00% $\pm$ 3.46 |

When $\alpha = 1$, optimization focuses entirely on the teacher, although the student may still improve through shared parameters. When $\alpha = 0$, training is focused on student learning from the teacher's current behavior. Where, we observe that under certain conditions, parameter sharing can still lead to improvements in the teacher without explicit teacher updates. When $\alpha = 0.5$, both are optimized jointly. Shared parameters allow representations learned for using PI to transfer to the student, while student updates keep those representations effective without PI. The full algorithm is given in Algorithm 1.

**Connection to Variational EM.** This approach can be viewed as a form of Variational Expectation-Maximization (EM), where one uses an approximate posterior $\pi^T$ to approximate a target distribution $\pi^*$. Here the E-step first improves $\pi^T$ and the M-step distills this into the student policy $\pi^S$. Traditionally this can be trained sequentially or in alternating loops with separate models (see Zhou et al. (2025)). We discuss this connection more in depth in App. E.1 and App. E.2, characterizing the target distribution $\pi^*$, derive $\pi$-Distill from this perspective and compare against sequential setups (similar to Zhou et al. (2025)).

### 3.2. On-Policy Self Distillation

**Motivation.** $\pi$-Distill can be viewed as off-policy learning, where the student is trained on trajectories generated by the PI-conditioned teacher. A complementary line of work

studies on-policy distillation, in which the student acts as the sampling policy and knowledge is transferred by minimizing the reverse KL between the student and teacher (Agarwal et al., 2024; Yang et al., 2026). Prior work typically utilizes a larger model as the teacher. We introduce this objective in our PI setting by instantiating it with the same shared-parameter model, where the teacher is additionally conditioned on PI. We refer to this objective as On-Policy Self-Distillation (OPSD).

**Algorithm.** The above intuition yields the following objective:

$$J_{\text{OPSD}}(\theta) = \mathbb{E}_{\substack{\mathbf{o} \sim \pi_\theta^{\text{S}}(\mathbf{o}|s) \\ \mathbf{s} \sim P}} \Big[ R(\mathbf{o}, \mathbf{s}) \Big] - \beta \, D_{\text{KL}} \big( \pi_\theta^{\text{S}}(\mathbf{o}|s) \; \| \; \text{sg}(\pi_\theta^{\text{T}}(\mathbf{o} \mid s, \mathbf{I})) \big).$$

Note that the updates are on-policy as the expectation is taken over $\pi^{\text{S}}$. Where the reverse KL acts as a dense per-token reward measuring how closely the student matches the teacher. The full algorithm is given in Algorithm 2. We analyze OPSD in greater depth in App F and characterize the specific target distribution that the algorithm implicitly fits. Concurrent work also propose this objective, demonstrating its effectiveness in settings such as having access to ground truth answers (Zhao et al., 2026), conditioning on reflective self-feedback (Hübotter et al., 2026), and continual learning (Shenfeld et al., 2026). In this work, we introduce and evaluate OPSD for PI transfer without ground truth and characterize it across a variety of PI types, identifying settings in which it fails.

# 4. Experimental Setting

While learning with PI is applicable to many settings, we ground our work in the task of distilling frontier models within multi-turn tool-calling environments. We focus on this domain for two primary reasons. First, non-frontier models often lack the capabilities required for such settings, whereas frontier models demonstrate proficient performance (Singh et al., 2025; Team et al., 2025). Consequently, weaker models struggle to independently sample successful trajectories, making the PI derived from a frontier model's actions highly valuable. Second, frontier models typically occlude their CoT reasoning (OpenAI et al., 2024). This renders standard distillation methods infeasible, a gap our proposed algorithms are designed to fill.

## 4.1. Benchmarks

To evaluate our approach, we first employ $\tau$-Bench (Yao et al., 2024), which simulates customer service interactions where agents book flights (airline domain) or assist shoppers (retail domain) by calling tools and gathering user information. To reduce computational costs, we substitute the GPT-4o user simulator with Qwen-14B. The most substantial change is that we remove the

`transfer_to_human_agents` tool, as it consistently led to reward hacking. The resulting dataset consists of 500 training tasks restricted to the retail domain; we evaluate on 115 held-out retail tasks and 50 airline tasks, utilizing the latter to test Out-Of-Domain (OOD) generalization. Next, we utilize Travel Planner (Xie et al., 2024), a benchmark focusing on tool use for planning. While the original repository[‡] employs a rubric-based evaluation that prioritizes "easy" constraints before checking "hard" ones, we found this setup causes policies to collapse onto undesired behaviors (see App. K). To address this, we decouple the rewards so that easy constraints are tied directly to their corresponding hard constraints, for example, verifying dietary restrictions immediately after booking a restaurant. We train on the 45 training tasks and report results on the 180 publicly available held-out tasks. Finally, to probe whether training on these environments enhances tool usage to scopes far beyond the training domains, we evaluate on the GEM QA multi-turn tool-usage environment suite (Liu et al., 2025b). This suite equips agents with a search tool consisting of seven environments (2Wiki (Ho et al., 2020), PopQA (Mallen et al., 2022), TriviaQA (Joshi et al., 2017), HotpotQA (Yang et al., 2018), Bamboogle (Press et al., 2023), NaturalQuestions (Kwiatkowski et al., 2019), and Musique (Trivedi et al., 2022)).

## 4.2. Sources of Privileged Information

To show the benefits of leveraging train-time PI, we aim to distill a frontier model using only its raw output trajectories. For this, we mine trajectories from `DeepSeek-chat-v3.1` as it is open-source and allows access to its reasoning tokens. Having access to these tokens allows us to benchmark against standard settings that assume access to CoT. This is an important detail, as baselines that assume access to this information should act as a *soft*-upper bound.

**Types of Privileged Information.** We transform the raw trajectories into three distinct forms of PI to analyze how varying information density impacts performance.

1. **Tool calls & arguments.** The complete sequence of actions executed by the frontier model, including function names and input arguments, e.g. `GetUserDetails(Name:"Kevin Lau")`.

2. **Tool calls only.** We provide only the function names like `GetUserDetails` and require the model to infer the correct arguments from the context.

3. **Self-generated hints.** We prompt the *trained* model to summarize a successful trajectory produced by the frontier model. This process can help filter inefficient actions and

---

[‡]https://github.com/OSU-NLP-Group/TravelPlanner

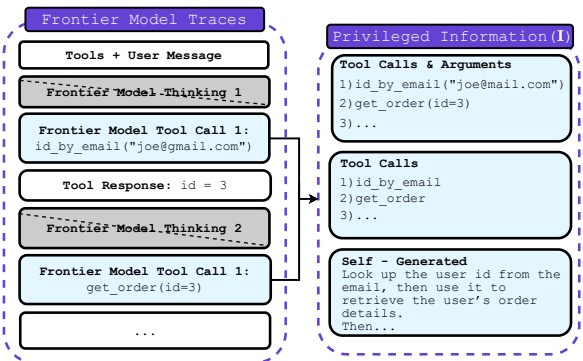

*Figure 3.* **Deriving PI from frontier model traces.** The left panel illustrates sampling trajectories from a frontier model, where full CoT reasoning is occluded (OpenAI et al., 2024). We transform these raw traces into three types of PI with varying information density: (1) **Tool Calls & Arguments**, retaining the exact actions invoked by the frontier model; (2) **Tool Calls Only**, where arguments are stripped; and (3) **Self-Generated Hints**, where the student model summarizes the frontier trajectory into a concise hint.

suggest ways to improve upon the expert's behavior (see App. L.3 for an example).

Leveraging these three formats enables us to identify what properties allow successful transfer between teacher and student. We analyze this in Section 7, finding that the optimal configuration depends on the value of $\alpha$ and the specific attributes of the PI (e.g. KL between teacher and student).

Using this setup, we collect 15,885 successful traces for $\tau$-Bench retail and 1,986 for Travel Planner, evenly sampled across tasks. From these traces, for each training task, we select the successful trajectory with the least number of steps to build the PI. In total, we obtain PI for all 45 tasks in Travel Planner and 300/500 tasks in $\tau$-Bench. We note we exclude the self-generated hints for R1-Deepseek-Distill-LLama-8B as it consistently returns the raw input trace or tool calls as the hint.

### 4.3. Models and Baselines

**Models.** We employ Qwen3-4B and Qwen3-8B (Yang et al., 2025a) being strong reasoning models. We also evaluate R1-Distill-Llama-8B to cover a distinct model family. We find this model fails to generate correct trajectories even when conditioned on PI in both benchmarks, making direct RL training unfeasible. Thus, exclusively for R1-Distill-Llama-8B, we warm-start it using SFT w/ CoT from expert traces. This setup allows us to determine if PI remains beneficial even when the model has already seen it during training.

**Baselines.** We instantiate $\pi$-Distill with $\alpha \in \{0, 0.5, 1\}$ and OPSD. We compare against (i) standard RL, (ii) SFT

on expert trajectories with and without CoT, and (iii) SFT followed by RL. We utilize the GRPO objective outlined in Section 2 for all RL-based algorithms. Following Vattikonda et al. (2025), we sweep over multiple SFT checkpoints for SFT+RL baselines and report results using the checkpoint that yields the strongest final performance. We use the full set of collected successful traces for all SFT baselines, as this maximized performance. For $\pi$-Distill and OPSD in $\tau$-Bench, we utilize PI whenever available otherwise, we sample the traces with the student and perform regular RL for that goal.

**Implementation Details.** We run all experiments using 2 H100 GPUs with a context limit of 25k tokens (with one exception, see App. I). We found initially that traces often exceeded this limit, thus adopting a length penalty reward that penalizes trajectories exceeding 15k tokens (full details in App. I.2). Additionally, we observed during training that a few tokens referencing the PI consistently exhibited very high KL (e.g., the token "hint"). We thus incorporate a penalty on the frequency of these tokens. Our final experiments incorporate this penalty, though we found in practice it makes little difference for final performance (see App. H for full details). All PI is added to the task description in the system prompt, using the prompts in App. L.4. We thoroughly sweep over relevant HPs for all baselines, for a full list see App. I.3. Using the best performing set, we run three seeds of this set and report their average. In experiments, we score models using the best running average over three subsequent checkpoints, capturing both performance and stability.

## 5. Main Results

In this section, we demonstrate that leveraging PI provides a potent learning signal on held-out tasks, proving effective even when full CoT supervision is available. Table 1 details performance metrics for Travel Planner and $\tau$-Bench. Our primary finding is that $\pi$-Distill variants achieve superior performance across all but a single setting. With Qwen3-4B we find that $\pi$-Distill using $\alpha = 0$ can outperform SFT + RL w/ CoT, with other variants approximating its performance. Notably, in Qwen3-8B, $\pi$-Distill consistently outperforms the industry standard (SFT w/ CoT + RL) regardless of $\alpha$. In the best-case scenarios, $\pi$-Distill achieves substantial improvements: 11.8% on Travel Planner, and 2.08% and 6.00% on the retail and airline subsets of $\tau$-Bench, respectively. These results are substantial as they confirm that $\pi$-Distill effectively distills frontier models even when CoT traces are hidden, enabling non-frontier models to become proficient at complex multi-step agentic tasks. Moreover, $\pi$-Distill achieves this with significantly greater efficiency. Unlike SFT w/ CoT + RL, which requires sweeping over multiple SFT checkpoints to achieve peak

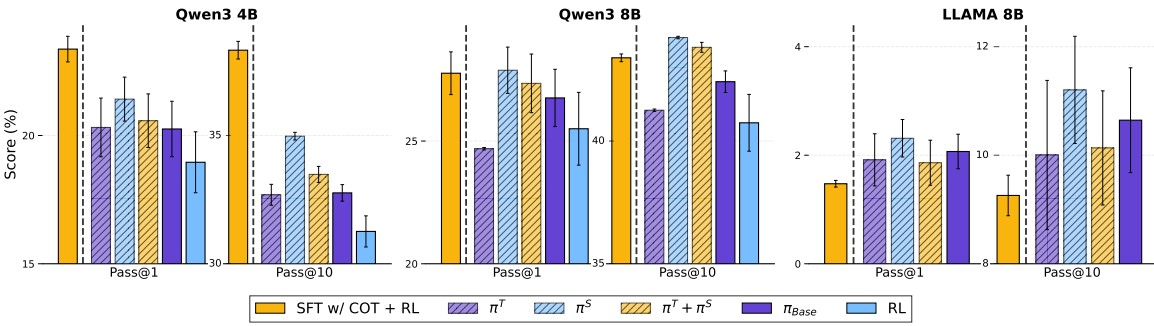

*Figure 4.* Pass@1 and Pass@10 on the GEM search-tool benchmark set (7 datasets) for Qwen3-4B, Qwen3-8B, and Llama-8B, using the best checkpoint selected on $\tau$-Bench Retail. Bars show mean $\pm$ SE over three seeds per dataset, comparing $\pi$-Distill variants ($\pi^T$, $\pi^S$, $\pi^T + \pi^S$) against SFT w/ CoT + RL, $\pi_{\text{Base}}$, and standard RL.

performance (Vattikonda et al., 2025), $\pi$-Distill requires only a single training phase, greatly simplifying the training process.

Regarding $\alpha$, we observe that there is no definitive best value, rather the best-performing $\alpha$ varies by setting. We analyze this nuance further in § 7. When analyzing other methods that do not assume access to full CoT, we find RL and SFT w/o CoT + RL perform similarly, while RL requires significantly less compute, but find both fail to yield substantial gains, significantly lagging behind SFT w/ CoT + RL. Additionally, we find that OPSD, can substantially outperform these baselines in $\tau$−Bench, while performing similarly in Travel Planner. We also observe that when OPSD succeeds, it provides substantial gains in OOD settings ($\tau$-Bench Airline), scaling with model capacity, being the second best in Qwen3-4B and best in Qwen3-8B. Finally, our results on R1-Distill-Llama-8B show that $\pi$-Distill remains useful even if the model has previously been SFTd on traces containing the PI.

## 6. Out of Domain Experiments (OOD)

In this section, we demonstrate that $\pi$-Distill generalizes effectively to OOD tasks, consistently outperforming standard RL and the base model, with this holding true for OPSD on Qwen3-8B. We report Pass@1 and Pass@10 metrics on the GEM (Liu et al., 2025b) search-tool benchmark suite which consists of 7 datasets. To simulate a realistic deployment scenario, we select the single best-performing checkpoint in $\tau$-Bench retail for each model. We then evaluate these checkpoints across the suite using three random seeds and report the aggregated mean and standard errors in Figure 4.

We find that for both Qwen3 models, $\pi$-Distill variants consistently outperform the base model and standard RL holding true for OPSD under more capable models. When compared to SFT w/ CoT + RL, we observe differences based on model size. On Qwen3-4B, SFT w/ CoT + RL is consistently the top performer with OPSD showing sig-

nificant degradation. For Qwen3-8B, however, we find that $\alpha = 0$ and $\alpha = 0.5$ variants of $\pi$-Distill as well as OPSD can significantly outperform SFT w/ CoT + RL. For $\pi$-Distill, this aligns with our findings in Section 5. Where for Qwen3-4B not all $\pi$-Distill variants show substantial improvements over SFT w/ CoT + RL, whereas for Qwen3-8B, the opposite trend is observed. We observe a similar trend for OPSD, where it shows significant degradation on Qwen3-4B, while on Qwen3-8B it shows significant improvements. We attribute this to the possibility that on smaller models, OPSD may overfit the teacher supervision for the task, while the more potent reasoners can provide more generalizable feedback. Overall, results on both $\pi$-Distill and OPSD imply that explicit supervision from frontier model's CoT is more important for smaller models, where stronger reasoners benefit from being more on-policy via self-generated CoT. Additionally, we observe deterioration in R1-Distill-Llama-8B under SFT w/ CoT + RL, where performance drops below the base model. While $\pi$-Distill and OPSD do not improve over the base model in this specific case, they avoid significant deterioration. Finally, raw RL consistently exhibits degradation relative to the base model across all evaluations. Overall, these results indicate that $\pi$-Distill is highly effective for distillation in OOD settings when CoT traces are occluded, preventing performance from regressing below the base model. Moreover, both $\pi$-Distill and OPSD show significant improvements as model size scales.

## 7. What Matters When Using Train-Time PI

While prior sections focused on the best-performing configurations, here we analyze how varying the type of PI affects final performance. Our goal is to isolate the factors that determine success when training with PI. We identify two primary drivers: (i) the divergence between the conditioned and unconditioned base policies ($D_{\text{KL}}(\|)$ for $\pi$-Distill and $D_{\text{KL}}(\|)$ for OPSD), and (ii) the usefulness of the privileged signal, captured by the utility

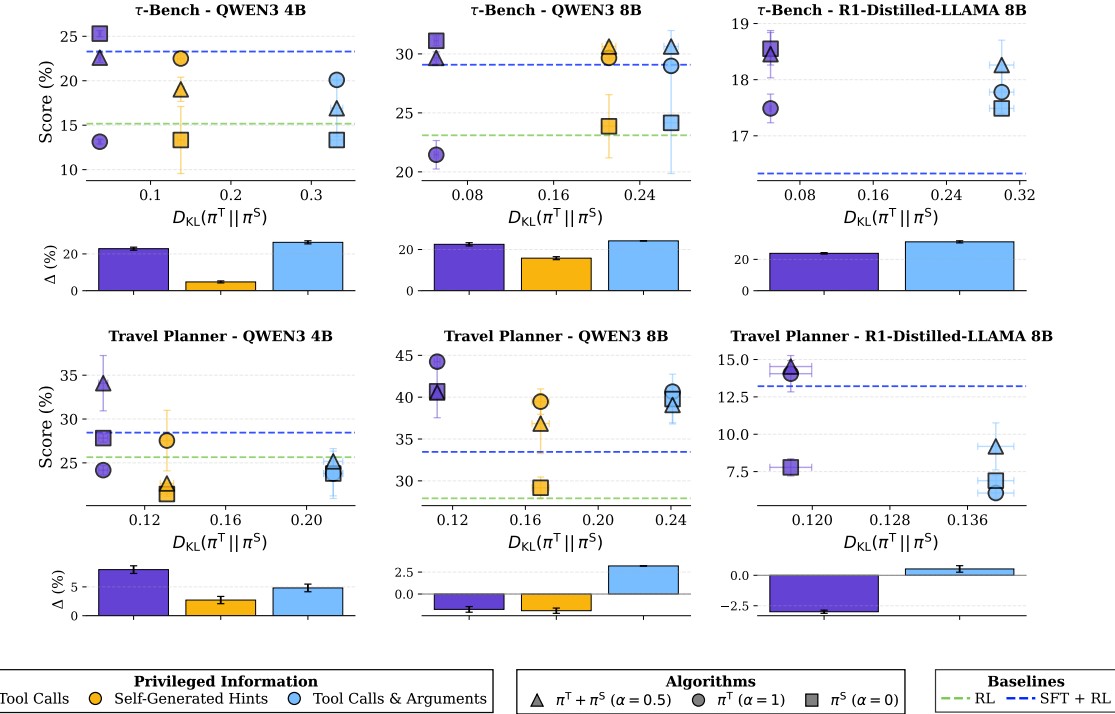

*Figure 5.* **Impact of PI Types and Algorithms on Performance**. We compare held-out performance on $\tau$-Bench (top row) and Travel Planner (bottom row) across three base models and three PI types (colors). The scatter plots map final scores against the initial teacher-student divergence ($D_{\mathrm{KL}}(\pi_{\mathrm{base}}^T \| \pi_{\mathrm{base}}^S)$), while the bar-charts display the PI utility ($\Delta$) on training tasks. Key observations: (1) Higher initial KL divergence generally correlates with decreased final performance. (2) Joint training ($\alpha = 0.5$, $\triangle$) is the most stable configuration, performing best in 6/16 scenarios and worst in a single one. (3) Student-only training ($\alpha = 0$, $\square$) requires low KL and positive utility (note the failure in Planner QWEN3 8B where $\Delta < 0$). Conversely, Teacher-only training ($\alpha = 1$, $\bigcirc$) degrades as KL increases or fails due to policy collapse when KL is negligible.

$\Delta = \mathrm{score}() - \mathrm{score}()$ on *training tasks*. Additionally, in Figure 6 we report the maximum attainable utility on $\Delta_{\max} = \max_t \mathrm{score}(\pi_t^{\mathrm{PI}}) - \max_t \mathrm{score}(\pi_t^{\mathrm{RL}})$, defined as the difference between the best scores on *training tasks* achieved with PI and without PI (pure RL), which measures how effectively each algorithm converts access to PI into performance gains.

### 7.1. What matters for $\pi$-Distill

Figure 5 displays held-out performance on both $\tau$-Bench Retail and Travel Planner across the types of PI defined in Section 4.2. Each subplot maps final performance against the initial divergence $D_{\mathrm{KL}}(\|)$, with accompanying bar plots showing the utility $\Delta$ for each type of PI. The most prominent pattern is that as the initial KL increases, final performance tends to decrease, though the full picture requires a more nuanced, $\alpha$-dependent analysis.

**Teacher only training $\alpha = 1$.** In this setting, we find performance generally declines or maintains as KL divergence increases. The primary exception occurs when using only tool calls in $\tau$-Bench, where this variant underperforms. We trace this failure mode to an early collapse in KL diver-

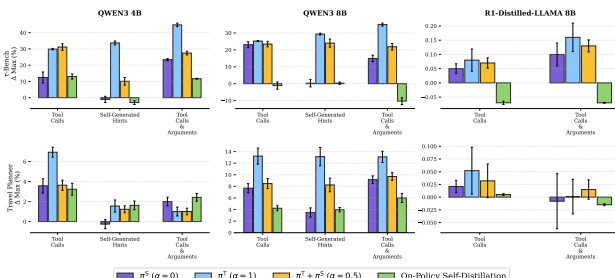

*Figure 6.* **Maximum Improvement ($\Delta_{\max}$) across PI Types and Algorithms.** We compare the peak performance gain over baselines for different PI variants (x-axis) and training configurations ($\alpha$) on TravelPlanner (bottom) and $\tau$-Bench (top). We find that PI types which initially underperform (e.g., self-generated hints on Planner, `Qwen3-8B`) can yield substantial gains when the teacher is trained to utilize them ($\alpha > 0$), confirming that learning to leverage PI is an important factor in transferring from teacher $\pi^{\mathrm{T}}$ to student $\pi^{\mathrm{S}}$.

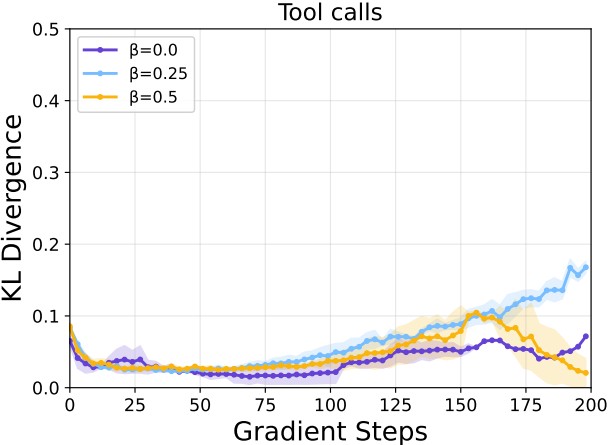

$\pi^T \mid$ KL Divergence During Training

*Figure 7.* **Training KL between $\pi^T$ and $\pi^S$ during training the teacher $\pi^S(\alpha = 1)$ on $\tau$-Bench.** We observe an early KL collapse making $\pi^S \approx \pi^T$. We attribute the underperformance of $\pi^T$ on low KL settings to this collapse.

gence. Figure 7 displays the KL as training progresses. We observe that even when $\beta = 0$ the KL drops to near zero, indicating teacher and student have collapsed onto each other ($\pi^T \approx \pi^S$). As a consequence, we find the teacher $\pi^T$ learns to ignore the P,I causing it to underperform even the RL baseline. We attempted to mitigate this by using $\pi_{\text{base}}$ instead of $\pi^S$ when calculating the KL, but found this further degraded results (see App. H.2).

In addition, our results confirm that training the teacher is an important part of leveraging PI. We can see this from the observation that even when the initial utility is negative ($\Delta < 0$), training the teacher allows the policy to learn to leverage the PI, consistently showing positive $\Delta_{\max}$ values in such cases. We find that effectively learning to use the PI is a significant contributing factor to transfer when only training the teacher.

**Joint training $\alpha = 0.5$.** We identify $\alpha = 0.5$ as the most robust configuration. It achieves the best performance in 7 out of 16 scenarios and effectively avoids the failure modes of the other variants, ranking as the worst performer only once. By balancing both teacher and student objectives, $\alpha = 0.5$ is able to mitigate the failure cases of independent training. We believe more granular tuning of $\alpha$ can likely lead to optimal performance in most settings, leaving this as future work.

**Student only training $\alpha = 0$.** Here, low KL divergence is a strong predictor of success. In $\tau$-Bench, for example, setting $\alpha = 0$ with only tool calls consistently yields the best results, as the minimal distribution shift makes learning from the conditioned traces significantly easier. As KL divergence rises, performance generally drops, though high-

utility PI can occasionally reverse this trend.

Further, we find that PI utility $\Delta$ can play a large role in the success of this variant. For instance, on Travel Planner with `Qwen3-8B`, $\alpha = 0$ underperforms significantly because the PI provides negative utility ($\Delta < 0$), actively degrading performance. Furthermore, as seen in Figure 6, we find that $\Delta_{\max} \approx 0$ in these failure cases, confirming that the algorithm cannot extract value from the PI when the signal itself offers no initial advantage over the base model. On the other hand, we find that when the teacher-student KL is low, student-only training effectively transfers knowledge back onto the teacher. This allows the teacher to learn how to leverage PI even though it is not being directly trained. We observe this behavior in `Qwen3-8B` on Travel Planner when using only tool calls as PI. While the initial $\Delta < 0$, the low KL allows transfer to occur from the student to the teacher and enables the teacher to sample improved traces.

## 8. Limitations & Future Work

While we show that both $\pi$-Distill and OPSD are effective algorithms when CoT is not available, all our PI is transformed from frontier model traces. Exploring how to efficiently obtain useful PI when neither frontier model actions nor ground-truth answers are available is an interesting direction. Concurrent work by Hübotter et al. (2026) explores this direction by mining PI through self-reflection. Additionally, our experiments are limited to models with $\leq 8B$ parameters. Further scaling of both $\pi$-Distill and OPSD can shine light on additional important factors to enable these algorithms. Moreover, our analysis in Section 7 is limited to observational studies, where we do not systematically control for all variables, but rather observe properties of different configurations. A study that systematically controls for these factors could help in mitigating failure cases and obtaining the most useful PI.

## 9. Conclusion

We introduce $\pi$-Distill and On-Policy Self-Distillation, two algorithms that leverage PI at training time to produce an improved policy even when that information is lacking at test time. By grounding our work in the distillation of frontier models where CoT reasoning is inaccessible, we demonstrate that the absence of these proprietary traces is not a limiting factor. In fact, our approach remains effective even when compared to industry standards that assume access to full CoT reasoning. Through evaluations across two training domains and eight OOD datasets, we find that $\pi$-Distill and OPSD are highly effective in all explored settings. Finally, we characterize the factors driving the distillation of training-time PI, showing that its success can often be predicted using only base-model statistics.

## Impact Statement

This work proposes and evaluates privileged-information distillation for language-model agents, where extra training-time signals (such tool-call traces or hints) help learning while test-time operation must act without those signals. The primary positive impact is improving sample efficiency and reducing the compute needed to obtain capable tool-using agents, which can lower cost and energy and make stronger agents accessible on smaller models. Better tool-use can also benefit benign applications that rely on structured actions, such as information access, workflow automation, and planning in constrained environments.

There are also potential risks. Increasing the capability of tool-using agents can enable misuse through automation at scale or over-reliance in settings where mistakes have real consequences. In addition, the use of privileged signals can lead models to explicitly mention or reproduce hint-related content in their outputs during training and evaluation, which may be undesirable in deployment. We therefore measure privileged-information leakage and study a simple penalty designed to discourage explicit mention of privileged content, and we recommend that future work develop stronger leakage controls and safety evaluations alongside capability gains.

Overall, our results suggest privileged-information distillation can improve performance and efficiency, but it should be paired with careful evaluation and deployment safeguards that limit harmful actions and discourage leakage-like behaviors.

## Acknowledgments

We thank and credit Siddarth Venkatraman for the idea to view learning from PI from a variational perspective, which led to the final version of $\pi$-Distill. We thank Vedant Shah for helpful discussion and proof reading initial drafts of the paper. Additionally, we thank Michael Noukhovitch for valuable feedback that substantially aided the final draft of this work. We also thank Alexandre Piché for their valuable discussions.

EP acknowledges the support of the NSERC PGS-D grant, the Bourse en intelligence artificielle provided by Université de Montréal and the MITACS Accelerate grant. DV acknowledges the support of the MITACS Accelerate grant. LC recognizes the support of NSERC, the Canada CIFAR AI Chair Program, the Canada First Research Excellence Fund and IVADO.

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

# A. Author Contributions

**Emiliano Penaloza** proposed the idea of leveraging PI, led the project, implemented all algorithms and the RL environments, conducted most experiments, and was the primary writer of the paper.

**Dheeraj Vattikonda** was a core part of the project, with the final version heavily improved by their contributions. Specifically, they provided substantial support on all experiments over the course of the project, implemented and conducted experiments on OOD environments, and provided heavy contributions to paper writing and figures.

**Nicolas Gontier** helped in advising the project and providing detailed feedback that substantially improved experimental rigor.

**Alexandre Lacoste** advised the project, providing valuable insights to the goal and purpose of leveraging PI, helped improve experimental rigor and analysis, and assisted in paper writing.

**Laurent Charlin** advised the project, helping influence the framing and perspective of the project, and provided help with writing, with a substantial portion of the introduction written by them.

**Massimo Caccia** was a core member of the project. They helped shape the research direction, proposed the on-policy self-distillation component, guided the design of extensive ablation studies, and provided detailed technical guidance across all stages. They also contributed substantially to writing and figures. (They are also responsible for the green outline in the abstract when it should have been purple.)

# B. Related Work

**Latent Reasoning and Variational Perspectives.** Recent work increasingly frames LM reasoning as a latent-variable inference problem (Hu et al., 2024; Sordoni et al., 2023; Luo et al., 2025; Li et al., 2025). Within this framework, the most popular approach is STaR (Li et al., 2025), which uses a hint when the model is unable to correctly answer the question and then uses SFT to fit onto the generated reasoning trace. This is similar to student-only training in $\pi$-Distill ($\alpha = 0$), where, rather than a KL-regularized off-policy RL objective, SFT is used, and the teacher is not trained. Most similar to $\pi$-Distill is the work by (Zhou et al., 2025), which proposes variational reasoners. This approach can be seen as a modified version of STaR, where both teacher and student are being trained iteratively. The main difference between variational reasoning and our work is that they assume access to oracle answers and perform an iterative version of variational EM using separate parameters for the teacher and student. In comparison, we simplify the training objective by allowing the teacher and student to share parameters and do not assume access to ground truth answers. We compare against a similar setup with minor modifications due to not having access to ground truth solutions in App. E.2.

**Self-Bootstrapping, Privileged Signals, and Guided Exploration.** Complementary work focuses on using privileged or auxiliary signals to enable learning in hard regimes. (Chen et al., 2025) injects self-generated high-level hints into online RL to overcome zero-reward exploration barriers, while (Qu et al., 2026) uses privileged oracle solutions as structured on-policy exploration signals for hard reasoning tasks. Liu et al. (2026) use self-generated memory as an exploration scaffold and combine on- and off-policy updates to distill memory-guided behavior into a policy that remains effective without memory. Both these algorithms can be viewed as training the teacher to use PI with implicit transfer via parameter sharing. Additionally, other lines of work explore distilling certain skills/behaviors in the model via contextualized sampling (Didolkar et al., 2025; 2024; Yang et al., 2025b; Qu et al., 2025). We believe our proposed methods could be used to distill desired skills/behaviors into model weights in a more effective manner compared to traditional SFT.

**Context Distillation.** Both our proposed methods can be seen as a form of context distillation. This growing line of work studies how to train models to internalize context-dependent reasoning so that the benefits of rich contexts (e.g., instructions, demonstrations, auxiliary computations) are distilled into the model's weights (Caccia et al., 2025; Snell et al., 2022; Huang et al., 2022; Cao et al., 2025). Other works analyze this problem by performing SFT answers they give when having access to context (Snell et al., 2022). Moreover, other works also try to encapsulate such knowledge into adapters that can be plugged and played at test time (Caccia et al., 2025).

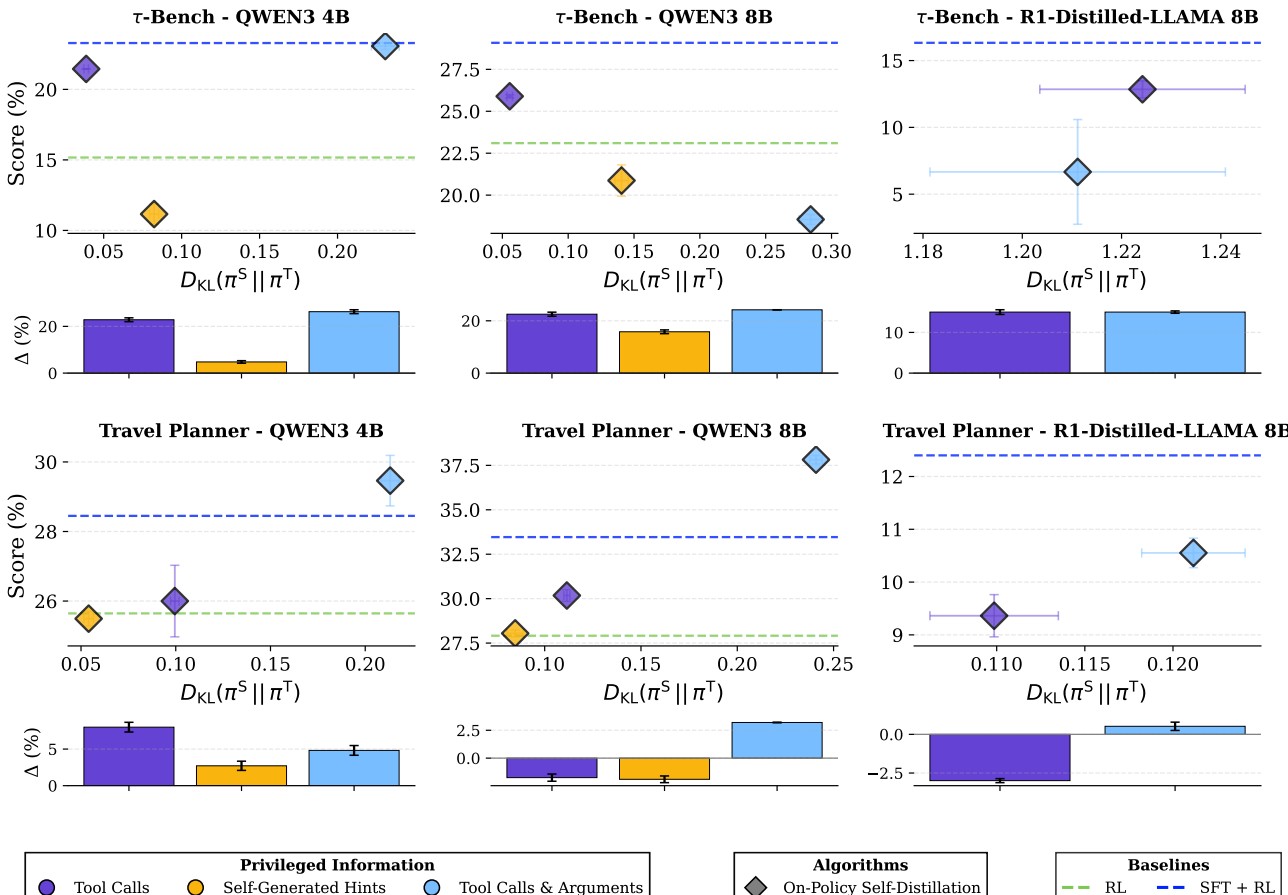

*Figure 8.* **Performance and Stability Analysis of OPSD.** We compare held-out performance on $\tau$-Bench (top row) and Travel Planner (bottom row) across three base models and three PI types (colors). The scatter plots map final scores against the student-teacher KL divergence ($D_{\mathrm{KL}}(\pi^S \| \pi^T)$), while the bar-charts display the PI utility ($\Delta$). Key observations: (1) Unlike $\pi$-Distill, higher KL are not always detrimental, rather, information richness of PI is most important, finding (**Tool Calls & Arguments**) often performs best (e.g., all results on Travel Planner and `Qwen3-4B` on $\tau$-Bench ). (2) Excessive KL can override positive utility (note `Qwen3-8B` on $\tau$-Bench, where $\Delta > 0$ but the high KL degrades performance). (3) `R1-Distill-Llama-8B` consistently struggles, which we attribute to either extreme KL divergence ($\tau$-Bench) or negative PI utility (TravelPlanner).

## C. What Matters for OPSD

Figure 8 similarly to Figure 5 displays performance on $\tau$-Bench retail and Travel Planner across PI types outlined in Section 4.2.

**Findings.** Our main finding is that, contrary to $\pi$-Distill, high-KL PI types do not necessarily indicate worse performance. Instead, the information content of the PI is the strongest predictor of successful transfer. For example, Tool Calls & Arguments, being the richest in information, consistently performs best for both `Qwen3` models on Travel Planner, and also achieves the best performance for `Qwen3-4B` on $\tau$-Bench.

Interestingly, although we find that high-information PI is generally best suited for transfer, we observe exceptions. In particular, for `Qwen3-8B` on $\tau$-Bench, Tool Calls & Arguments performs the worst and exhibits the highest KL. In this case, we observe that $\Delta_{\max}$ is negative, indicating that the reverse-KL penalty can inhibit training. Additionally, regarding the failure on `R1-Distill-Llama-8B`, while this could be due to the SFT phase prior to OPSD, we find that on $\tau$-Bench the KL for both PI types is exceptionally high. While on Travel Planner, the PI utility, $\Delta$ is low or negative, possibly explaining the diminished results. Overall, we find that the strongest predictor of performance for OPSD is the information content of the PI. While using the richest form of PI can lead to the best results, careful consideration is required to ensure the KL does not become excessive and hinder training.

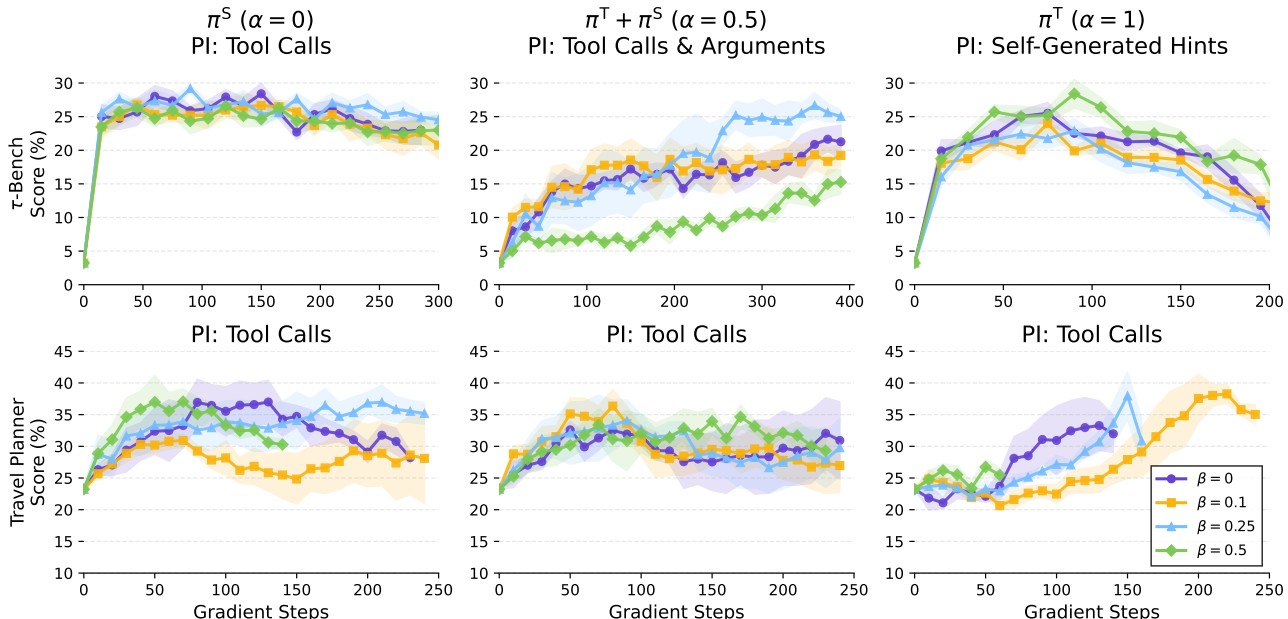

*Figure 9.* **Evaluation performance throughout training for $\pi$-Distill variants across varying KL penalties** ($\beta$). Runs that deteriorate significantly early are truncated for visual clarity. We observe that for settings involving teacher updates ($\alpha > 0$), a non-zero penalty ($\beta > 0$) is crucial for stabilizing training and achieving peak performance. Here error bars indicate standard errors. Discrepancies between plot and table values are addressed in App. J

## D. Ablation on $\beta$

In this section, we analyze $\beta$, the term controlling the regularization between the student and teacher, finding it important for achieving the best performance in 17/21 ablated configurations. We conduct extensive ablations on $\beta$ keeping all other parameters fixed, sweeping over $\beta = \{0, 0.1, 0.25, 0.5\}$. We analyze *all* values of $\alpha$ and PI types for both $\pi$-Distill and OPSD on $\tau$-Bench using both `Qwen3` models, as well as the best-performing PI type for Travel Planner. All results are reported over three random seeds.

Figure 9 shows a subset of the learning curves on held-out data for the best-performing PI type and $\alpha$ value in $\tau$-Bench Retail and Travel Planner, while Figure 16 provides the full set for $\tau$-Bench. For $\pi$-Distill, we consistently find that $\beta > 0$ aids in obtaining the best performance, particularly when the teacher is being trained ($\alpha > 0$). While $\beta$ can be sensitive, with no single value being the best in all settings, setting $\beta$ higher than 0 generally allows for better or matching performance, with only four cases showing inferior results. For OPSD, we find that the value of $\beta$ is less important with our results on Section 7 showing that information granularity and student-teacher KL are more important factors.

## E. Connections To Variational EM

In this Appendix, we derive $J_{\pi-\text{Distill}}$ using a variational perspective. Specifically, we show how $J_{\pi-\text{Distill}}$ can be seen as a joint EM algorithm where E and M are done simultaneously.

Traditional variational-EM aims to fit a parameterized distribution $\pi^{\text{S}}$ onto a target posterior $\pi^*$. An assumption this framework makes is that using $\pi^{\text{S}}$ to approximate $\pi^*$ may be suboptimal, rather, leveraging an approximate posterior $\pi_\theta^{\text{T}}(\mathbf{o} \mid s, \mathbf{I})$ conditioned on additional information can make the approximation easier. This is true when $\pi^{\text{T}}$ can sample from high-reward modes that $\pi^{\text{S}}$ cannot, therefore making optimization via RL impossible for $\pi^{\text{S}}$. We visualize the procedure in Figure 10

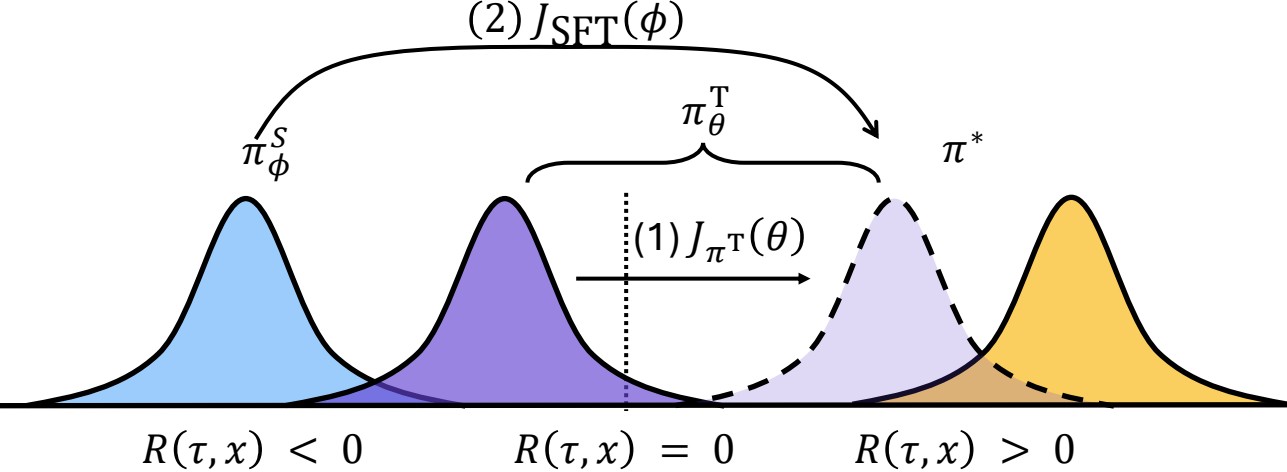

*Figure 10.* **Illustration of the Variational EM procedure for policy optimization**. The initial student policy, $\pi_\theta^S$, lacks support over trajectories with positive rewards ($R(\mathbf{o}, \mathbf{s}) > 0$), preventing direct improvement. To address this: (1) the teacher policy $\pi_\theta^T$ is optimized via $J_{\text{Teacher}}(\theta)$ to sample successful traces and approximate the optimal policy $\pi^*$; (2) the student policy is then updated via $J_{\text{SFT}}(\theta)$ to distill the knowledge from the teacher. While this two-step procedure is principled, it is computationally inefficient due to the requirement of maintaining dual parameter sets and distinct training phases. In contrast, $\pi$-Distill simplifies this pipeline into a single-phase process, providing superior performance with reduced complexity.

### E.1. Variational EM

**E-step $J_{\text{Teacher}}$.** We first define the target posterior we want to fit, $\pi^*$, as a reward-tilted posterior relative to the reference policy $\pi_{\text{ref}}$. For a given state $\mathbf{s}$, we define:

$$\pi^*(\mathbf{o} \mid \mathbf{s}) = \frac{\pi_{\text{ref}}(\mathbf{o} \mid \mathbf{s}) \exp(R(\mathbf{o}, \mathbf{s}))}{Z} \tag{1}$$

where $Z = \sum_{\mathbf{o}'} \pi_{\text{ref}}(\mathbf{o}' \mid \mathbf{s}_0) \exp(R(\mathbf{o}', \mathbf{s}))$ is the partition function and $\mathbf{o} = (\mathbf{z}, \mathbf{a})$ consists of internal reasoning tokens $\mathbf{z}$ and action tokens $\mathbf{a}$.

The partition function makes this distribution intractable, but it can be approximated using a variational posterior $\pi_\theta^T(\mathbf{o} \mid s, \mathbf{I})$ conditioned on privileged information $\mathbf{I}$. We optimize parameters $\theta$ by minimizing the reverse KL between the variational distribution and the target policy:

$$
\begin{aligned}
J_{\text{Teacher}}(\theta) &= -D_{\text{KL}}\big(\pi_\theta^T(\mathbf{o} \mid s, \mathbf{I}) \parallel \pi^*(\cdot \mid \mathbf{s})\big) \\
&= -\mathbb{E}_{\substack{\mathbf{o} \sim \pi_\theta^T(\mathbf{o}|s,\mathbf{I}) \\ \mathbf{s} \sim P}} \left[\log \frac{\pi_\theta^T(\mathbf{o} \mid s, \mathbf{I})}{\pi^*(\mathbf{o} \mid \mathbf{s})}\right] \\
&= \mathbb{E}_{\substack{\mathbf{o} \sim \pi_\theta^T(\mathbf{o}|s,\mathbf{I}) \\ \mathbf{s} \sim P}} \left[\log \pi^*(\mathbf{o} \mid \mathbf{s}) - \log \pi_\theta^T(\mathbf{o} \mid s, \mathbf{I})\right]
\end{aligned} \tag{2}
$$

Substituting the definition of $\pi^*$ into the equation:

$$
\begin{aligned}
J_{\text{Teacher}}(\theta) &= \mathbb{E}_{\substack{\mathbf{o} \sim \pi_\theta^T(\mathbf{o}|s,\mathbf{I}) \\ \mathbf{s} \sim P}} \left[\log \left(\frac{\pi_{\text{ref}}(\mathbf{o} \mid \mathbf{s}) \exp(R(\mathbf{o}, \mathbf{s}))}{Z}\right) - \log \pi_\theta^T(\mathbf{o} \mid s, \mathbf{I})\right] \\
&= \mathbb{E}_{\substack{\mathbf{o} \sim \pi_\theta^T(\mathbf{o}|s,\mathbf{I}) \\ \mathbf{s} \sim P}} \left[R(\mathbf{o}, \mathbf{s}) + \log \pi_{\text{ref}}(\mathbf{o} \mid \mathbf{s}) - \log \pi_\theta^T(\mathbf{o} \mid s, \mathbf{I}) - \log Z\right] \\
&= \mathbb{E}_{\substack{\mathbf{o} \sim \pi_\theta^T(\mathbf{o}|s,\mathbf{I}) \\ \mathbf{s} \sim P}} \left[R(\mathbf{o}, \mathbf{s}) - \big(\log \pi_\theta^T(\mathbf{o} \mid s, \mathbf{I}) - \log \pi_{\text{ref}}(\mathbf{o} \mid \mathbf{s})\big)\right] - \log Z \\
&\propto \mathbb{E}_{\substack{\mathbf{o} \sim \pi_\theta^T(\mathbf{o}|s,\mathbf{I}) \\ \mathbf{s} \sim P}} [R(\mathbf{o}, \mathbf{s})] - D_{\text{KL}}\big(\pi_\theta^T(\mathbf{o} \mid s, \mathbf{I}) \parallel \pi_{\text{ref}}(\cdot \mid \mathbf{s})\big)
\end{aligned} \tag{3}
$$

Since $\log Z$ depends only on the context $\mathbf{s}$, it is constant with respect to $\theta$ and can be omitted.

**M-step** $J_{\text{SFT}}$. If $\pi^*$ were a tractable distribution, we could directly fit the target policy by minimizing the forward KL divergence, $D_{\text{KL}}(\pi^* \parallel \pi_\theta^{\text{S}})$, corresponding to standard SFT. Since $\pi^*$ is intractable, we instead substitute it with our learned approximation $\pi_\theta^{\text{T}}$, yielding the following objective:

$$J_{\text{SFT}}(\theta) = \mathbb{E}_{\substack{\mathbf{o} \sim \pi_\theta^{\text{T}}(\mathbf{o}|s,\mathbf{I}) \\ \mathbf{s} \sim P}} \left[ \log \pi_\theta^{\text{S}}(\mathbf{o}|s) \right]. \tag{4}$$

Traditionally, one would either first fit $\pi_\theta^{\text{T}}$ to convergence and use it to fit $\pi_\theta^{\text{S}}$, or alternate between training the teacher and student. Regardless, this process requires two separate models and can be difficult to optimize. Our implementation is a modified version of Zhou et al. (2025), where the modifications are specific to our multi-turn setting, where no ground truth is provided.

### E.2. Failures of Variational EM.

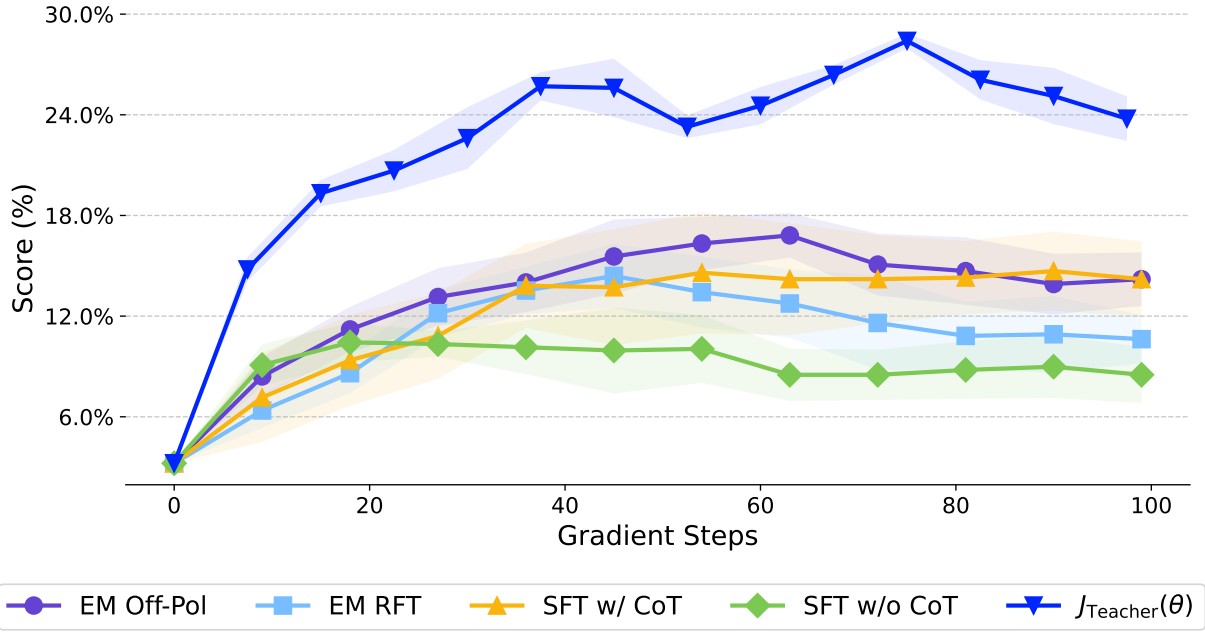

*Figure 11.* **Performance of Sequential EM variants on $\tau$-Bench Retail (`Qwen3-8B`).** We compare these methods against SFT baselines. While standard EM RFT (light blue) fails to match SFT w/ CoT (yellow), replacing RFT with off-policy RL (EM Off-Pol, purple) successfully allows for EM to outperform SFT w CoT. The strongest performance comes from $J_{\text{Teacher}}(\theta)$ (dark blue), demonstrating that parameter sharing between the teacher and student yields the most effective transfer. Shaded regions indicate standard error across 3 seeds.

Our initial efforts focused on making variational EM work, similar in spirit to (Zhou et al., 2025). We first document initial ideas that were unsuccessful, then we discuss the best version of EM we could find, being a sequential implementation. Finally, we discuss how these experiments ultimately motivated the design decisions for $\pi$-Distill.

**Initial Experiments**    First, we find that setting the reference policy $\pi_{\text{ref}}$ as a copy of the current student $\pi_\theta^{\text{S}}$ is crucial for learning. In contrast, using the fixed base model $\pi_{\text{base}}$ as a prior significantly hinders performance (see App. H.2).

Building on this insight, we initially explored an offline variational EM method similar to Zhou et al. (2025). Here, the goal is learning a CoT sampler as the variational posterior conditioned on expert traces. We do this with the following objective:

$$
J_{\text{Offline-EM}}(\theta) = \mathbb{E}_{\mathbf{s} \sim \mathcal{D}} \left[ \log \sum_{\mathbf{z}} \pi_\theta^{\text{S}}(\mathbf{a}, \mathbf{z} \mid \mathbf{s}) \exp(R(\mathbf{o}, \mathbf{s})) \right]
$$
$$
\propto \mathbb{E}_{\mathbf{a}, \mathbf{s} \sim \mathcal{D}, \mathbf{z} \sim \pi_\theta^{\text{T}}} \left[ R(\mathbf{o}, \mathbf{s}) + \log \pi_\theta^{\text{S}}(\mathbf{a} \mid \mathbf{z}, \mathbf{s}) \right] - D_{\text{KL}}\big(\pi_\theta^{\text{T}}(\mathbf{z} \mid \mathbf{s}) \parallel \pi_\theta^{\text{S}}(\mathbf{z} \mid \mathbf{s})\big) \tag{5}
$$

Notice here how the states are fixed and given in the offline dataset $\mathcal{D}$. Furthermore, this setup necessitates substantial hyperparameter tuning. Beyond the standard penalty $\beta$, the algorithm requires balancing the ratio of teacher-to-student updates, managing separate optimizer states, and calibrating the dataset size sampled from the teacher. Ultimately, we find this approach hard to implement and fails in our multi-turn setting. We attribute the success of Zhou et al. (2025) to their access to oracle answers; in our context, we find that fitting a variational posterior to suboptimal frontier model trajectories can even degrade performance.

Our following efforts focused on training the teacher online using Equation (2). We try this with alternating loops between student and teacher training, but find this to be ineffective. As we found it hard to avoid the student and teacher collapsing onto each other using this setup. Ultimately, we found a sequential version of EM worked best, though it still requires specific adjustments to achieve optimal performance.

**Sequential EM.**    We turn to Sequential EM after ruling out offline and alternating approaches. However, we find standard implementations remain unreliable for complex agentic tasks. Here, we analyze the limitations of standard EM and how they ultimately motivated $\pi$-Distill.

For this analysis, we use $\tau$-Bench Retail with `Qwen3-8B` using self-generated hints to train the teacher as this configuration performs best (see § 7). We compare sequential EM, where we first train the teacher and then perform SFT on successful trajectories (Rejection Fine-tuning or RFT), against SFT with and without CoT. We note that here it would be sufficient to outperform SFT without CoT as we assume no access to the internal CoT of the frontier model, thus in practice, sequential EM could replace the traditional SFT phase before RL.

We report results in Figure 11. First, we observe that standard Sequential EM outperforms naive SFT but lags behind training with CoT. We attribute the performance gap to the lack of negative feedback on failed trajectories. It only mimics the teacher. To address this, we replaced the RFT step with clipped off-policy RL. We find that this change enables sequential EM to outperform SFT with CoT.

Most surprisingly, in these experiments, we find that simply optimizing $J_{\text{Teacher}}(\theta)$ substantially improves the student $\pi_\theta^{\text{S}}$ when both teacher and student share parameters. This drastically outperforms all other baselines.

To summarize, our two main findings are that parameter sharing enables substantial transfer from teacher to student and the importance of leveraging negative traces via off-policy RL when training the student. We use these two findings to instantiate $\pi$-Distill while also simplifying the process from two training steps to a single one.

## F. On-Policy Self-Distillation $J_{\text{OPSD}}$

Contrary to the prior objectives, OPSD, rather, than trying to approximate $\pi^*$ can be framed as the reverse-Kl between $\pi$ and the target conditioned policy $\pi^{T*}$, being defined as:

$$
\pi^*(\mathbf{o} \mid \mathbf{s}, \mathbf{I}) = \frac{\pi_{\text{ref}}(\mathbf{o} \mid \mathbf{s}, \mathbf{I}) \exp(\frac{R(\mathbf{o}, \mathbf{s})}{\beta})}{Z^h} \tag{6}
$$

where $Z^h = \sum_{\mathbf{o}'} \pi_{\text{ref}}^T(\mathbf{o}' \mid \mathbf{s}, \mathbf{I}) \exp(R(\mathbf{o}', \mathbf{s}))$ is the partition function.

One can then approximate this distribution directly via reverse KL, giving the following objective:

To do this, we minimize the reverse KL between the student policy $\pi_\theta^S$ and the optimal privileged target distribution $\pi^*$:

$$
\begin{aligned}
D_{\text{KL}}(\pi_\theta^S \parallel \pi^*) &= \mathbb{E}_{\substack{\mathbf{o} \sim \pi_\theta^S(\mathbf{o}|s) \\ \mathbf{s} \sim P}} \left[ \log \frac{\pi_\theta^S(\mathbf{o} \mid \mathbf{s})}{\pi^*(\mathbf{o} \mid \mathbf{s}, \mathbf{I})} \right] \\
&= \mathbb{E}_{\substack{\mathbf{o} \sim \pi_\theta^S(\mathbf{o}|s) \\ \mathbf{s} \sim P}} \left[ \log \pi_\theta^S(\mathbf{o} \mid \mathbf{s}) - \log \left( \frac{\pi_{\text{ref}}(\mathbf{o} \mid \mathbf{s}, \mathbf{I}) \exp(R(\mathbf{o}, \mathbf{s})/\beta)}{Z^h} \right) \right] \\
&= \mathbb{E}_{\substack{\mathbf{o} \sim \pi_\theta^S(\mathbf{o}|s) \\ \mathbf{s} \sim P}} \left[ \log \pi_\theta^S(\mathbf{o} \mid \mathbf{s}) - \log \pi_{\text{ref}}(\mathbf{o} \mid \mathbf{s}, \mathbf{I}) - \frac{R(\mathbf{o}, \mathbf{s})}{\beta} + \log Z^h \right] \\
&= \mathbb{E}_{\substack{\mathbf{o} \sim \pi_\theta^S(\mathbf{o}|s) \\ \mathbf{s} \sim P}} \left[ -\frac{R(\mathbf{o}, \mathbf{s})}{\beta} + \log \frac{\pi_\theta^S(\mathbf{o} \mid \mathbf{s})}{\pi_{\text{ref}}(\mathbf{o} \mid \mathbf{s}, \mathbf{I})} \right] + \log Z^h \qquad (7) \\
&= -\frac{1}{\beta} \mathbb{E}_{\substack{\mathbf{o} \sim \pi_\theta^S(\mathbf{o}|s) \\ \mathbf{s} \sim P}} \left[ R(\mathbf{o}, \mathbf{s}) \right] + D_{\text{KL}}\left( \pi_\theta^S(\cdot \mid \mathbf{s}) \parallel \pi_{\text{ref}}(\cdot \mid \mathbf{s}, \mathbf{I}) \right) + \log Z^h. \qquad (8) \\
&\propto -\mathbb{E}_{\substack{\mathbf{o} \sim \pi_\theta^S(\mathbf{o}|s) \\ \mathbf{s} \sim P}} \left[ R(\mathbf{o}, \mathbf{s}) \right] + \beta D_{\text{KL}}\left( \pi_\theta^S(\cdot \mid \mathbf{s}) \parallel \pi_{\text{ref}}(\cdot \mid \mathbf{s}, \mathbf{I}) \right) \qquad (9)
\end{aligned}
$$

To align with $\pi$-Distill and the according to the results in App. H.2, we set $\pi_{\text{ref}} = \pi_\theta^T(\mathbf{o} \mid s, \mathbf{I})$

**Concurrent On-Policy Distillation Methods.** Concurrent work proposes closely related on-policy self-distillation frameworks. Recent methods train student models on their own trajectories while using privileged or conditioned variants of the same model as teachers, enabling on-policy transfer without off-policy distribution shift (Zhao et al., 2026; Shenfeld et al., 2026). These approaches are conceptually aligned with our OCPD formulation, but target supervised reasoning or continual learning settings, whereas our focus is agentic decision-making with training-time privileged information.

# G. Detailed Algorithms

Here we outline the detailed algorithms for both $\pi$-Distill and OPSD.

---

**Algorithm 1** $\pi$-Distill: Privileged Information Distillation for Language Models

---

1: **Input:** Dataset $\mathcal{D} = \{(s, \mathbf{I})\}$ where $\mathbf{I}$ is privileged information, Initial Policy $\pi_\theta$, Reference $\pi_{\text{ref}}$, $\alpha$, $\beta$, $\epsilon$, Learning rate $\eta$
2: **Initialize:** $\phi \leftarrow \theta$ {Parameters are shared between Teacher and Student}
3: **while** not converged **do**
4:     Sample batch $B = \{(s_i, \mathbf{I}_i)\}_{i=1}^N \sim \mathcal{D}$
5:     **// Step 1: Teacher Rollout (with Privileged Information)**
6:     **for** each $(s, \mathbf{I}) \in B$ **do**
7:         Sample $K$ trajectories $\{\mathbf{o}_1, \ldots, \mathbf{o}_K\} \sim \pi^T(\cdot \mid s, \mathbf{I})$
8:         Compute rewards $R(\mathbf{o}_k, s) = R_{\text{env}}(\mathbf{o}_k, s) - \beta D_{\text{KL}}[\pi^T(\cdot \mid s, \mathbf{I}) \| \pi_{\text{ref}}(\cdot \mid s, \mathbf{I})]$
9:     **end for**
10:    **// Step 2: Compute Group-Centered Advantages**
11:    **for** each $k \in \{1, \ldots, K\}$ **do**
12:       $\bar{R} = \frac{1}{K} \sum_{j=1}^K R(\mathbf{o}_j, s)$
13:       $A_k = R(\mathbf{o}_k, s) - \bar{R}$
14:    **end for**
15:    **// Step 3: Compute Objectives**
16:    **Teacher Objective (GRPO):**
17:    $J_{\text{Teacher}} = \frac{1}{K} \sum_{k=1}^K \min \left( \rho_k^{\text{teacher}} A_k, \text{clip}(\rho_k^{\text{teacher}}, 1 - \epsilon, 1 + \epsilon) A_k \right)$
18:    **Student Objective (Off-Policy GRPO):**
19:    Compute IS weights: $\rho_k = \frac{\pi^S(\mathbf{o}_k | \mathbf{s})}{\pi^T(\mathbf{o}_k | \mathbf{s}, \mathbf{I})}$ {Note: Student input is $s$ only}
20:    $J_{\text{Student}} = \frac{1}{K} \sum_{k=1}^K \min \left( \rho_k A_k, \text{clip}(\rho_k, 1 - \epsilon, 1 + \epsilon) A_k \right)$
21:    **// Step 4: Joint Update**
22:    $J_{\pi\text{-Distill}}(\theta) = \alpha J_{\text{Teacher}} + (1 - \alpha) J_{\text{Student}}$
23:    $\theta \leftarrow \theta + \eta \nabla_\theta J_{\pi\text{-Distill}}(\theta)$
24: **end while**

---

---

**Algorithm 2** On-Policy Self-Distillation

---

1: **Input:** Dataset $\mathcal{D} = \{(s, \mathbf{I})\}$ where $\mathbf{I}$ is privileged information, Initial Policy $\pi_\theta$, $\beta$, $\epsilon$, Learning rate $\eta$
2: **Initialize:** $\theta$ {Parameters for the Student Policy}
3: **while** not converged **do**
4:      Sample batch $B = \{(s_i, \mathbf{I}_i)\}_{i=1}^N \sim \mathcal{D}$
5:      **// Step 1: Student Rollout (On-Policy sampling)**
6:      **for** each $(s, \mathbf{I}) \in B$ **do**
7:          Sample $K$ trajectories $\{\mathbf{o}_1, \ldots, \mathbf{o}_K\} \sim \pi^S(\cdot \mid s)$
8:          {Reward Computation with Reverse KL}
9:          Compute rewards $R(\mathbf{o}_k, s) = R_{\text{env}}(\mathbf{o}_k, s) - \beta D_{\text{KL}} \left[ \pi^S(\cdot \mid s) \| \pi^T(\cdot \mid s, \mathbf{I}) \right]$
10:      **end for**
11:      **// Step 2: Compute Group-Centered Advantages**
12:      **for** each sample $i \in \{1, \ldots, N\}$ **do**
13:          $\bar{R}_i = \frac{1}{K} \sum_{j=1}^K R(\mathbf{o}_{i,j}, s_i)$
14:          **for** each $k \in \{1, \ldots, K\}$ **do**
15:              $A_{i,k} = R(\mathbf{o}_{i,k}, s_i) - \bar{R}_i$
16:          **end for**
17:      **end for**
18:      **// Step 3: Objective (Off-Policy GRPO)**
19:      Compute IS weights: $\rho_{i,k} = \frac{\pi_\theta^S(\mathbf{o}_{i,k}|s_i)}{\pi_{\text{old}}^S(\mathbf{o}_{i,k}|s_i)}$
20:      $J(\theta) = \frac{1}{K} \sum_{k=1}^K \min\left(\rho_{i,k} A_{i,k}, \text{clip}(\rho_{i,k}, 1-\epsilon, 1+\epsilon) A_{i,k}\right)$
21:      **// Step 4: Policy Update**
22:      $\theta \leftarrow \theta + \eta \nabla_\theta J(\theta)$
23: **end while**

---

# H. Additional Ablations

## H.1. Privilege Information leakage Ablation

We measure privileged-information leakage with a simple keyword detector applied to each assistant message, using the keyword list ["privileged information", "privileged info", "priv info", "secret information", "secret info", "correct tool calls", "secret", "privileged", "hint", "hints"]. For each occurrence, we assign a penalty of $-0.1$ and accumulate it over the trajectory, and we mark a trajectory as leaking if any of these appear. Fig 12 reports results at $\beta = 0.25$ and shows that enabling this leakage penalty does not change task performance in any noticeable way, since the learning curves with and without the penalty overlap closely. Fig 13 shows the corresponding leakage rate during training, and it rises as training progresses across all modes. The pattern with tool-calls showing the highest leakage, tool calls and arguments increasing more moderately, and self-generated hints starting lower but still increasing over time.

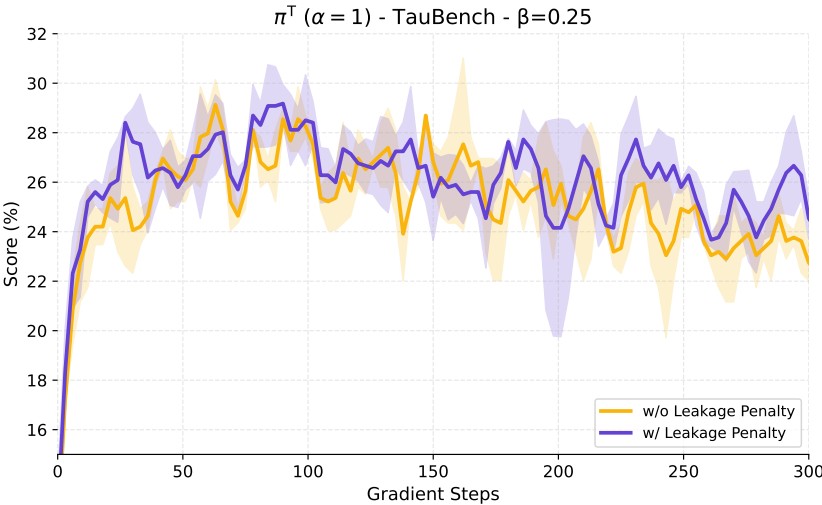

*Figure 12.* Performance with and without leakage penalty for $\pi^{\mathrm{T}}(\alpha = 1)$. We find that although the penalty does reduce the leakage of the privileged information (see Figure 13) it does not affect performance.

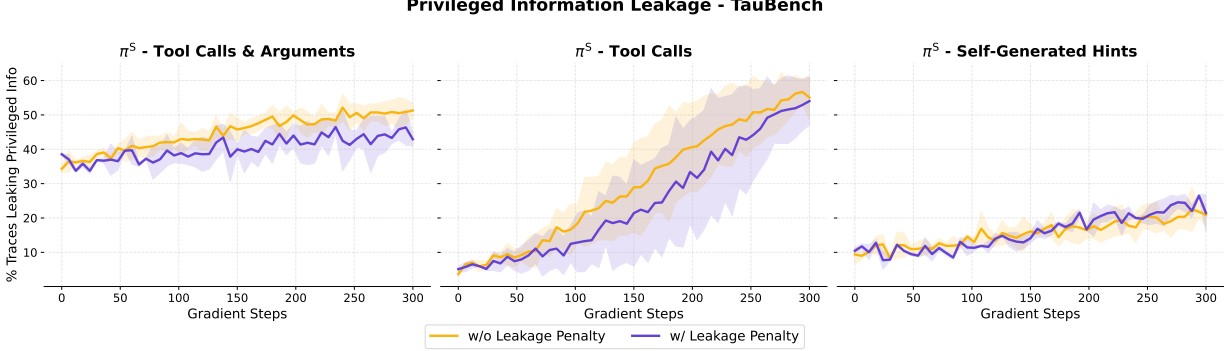

*Figure 13.* Proportion of traces leaking privileged information as training progresses, we see that regardless PI type, the leakage increases with more gradient steps, finding using a leakage penalty reduces this proportion, but not substantially.

**Test-time leakage.** As an additional analysis, we analyze whether leakage increases as models are trained with privileged data when evaluating $\pi^{\mathrm{S}}$. Figure 14 outlines this for `Qwen3-8B` for 300 gradient steps. We empirically validate that leakage does not significantly affect or increase when evaluating using $\pi^{\mathrm{S}}$ as the presence of leaked words does not meaningfully increase as training continues.

**Privileged Information Leakage in Evaluations - TauBench**

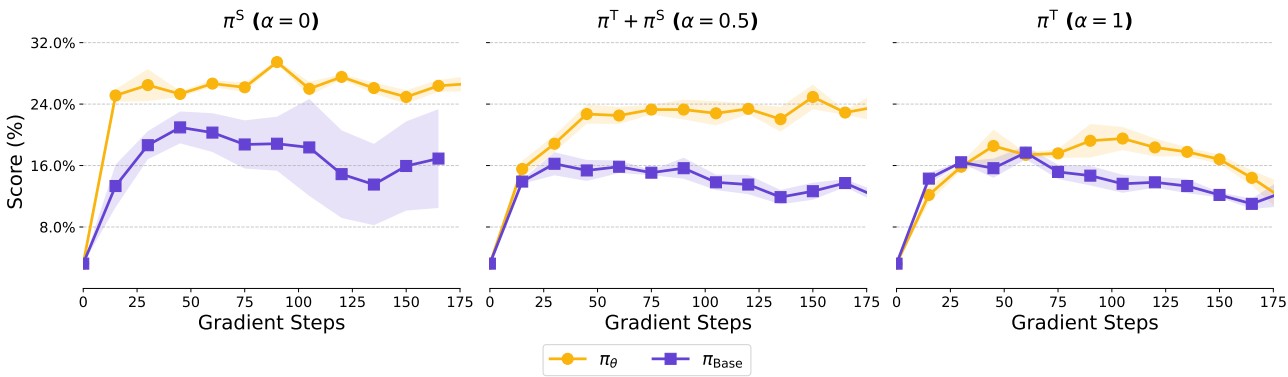

*Figure 14.* Proportion of traces leaking privileged information when **evaluating** $\pi^S$ as training progresses, we see that there is no increase in leakage when evaluated as training goes on.

## H.2. Reference Model Ablation

We ablate the choice of reference policy used in the KL term by comparing a fixed base reference, $\pi_{base}$, against using the student itself as the reference, $\pi_\theta$ (with stop-gradient on the reference branch). Fig.15 shows a clear difference in training stability. When $\pi_{base}$ is used as the reference, the performance degrades over training and can collapse most clearly in the $\pi^h$ setting ($\alpha = 1$), where the policy is pushed to move far from the base while still being penalized for that same deviation. In contrast, using $\pi_\theta$ as the reference (orange) yields stable learning across $\alpha \in \{0, 0.5, 1\}$, since the KL regularizer stays aligned with the current student distribution. Practically, this choice is also cheaper because using $\pi_\theta$ as the reference avoids maintaining a separate frozen reference model on the GPU, reducing memory and compute overhead.

*Figure 15.* Ablation between using reverse-Kl between the teacher and a prior. We ablate over two possible priors, $\pi_{base}$ and $\pi_\theta^S$. We find that using $\pi_\theta^S$ to be highly important in obtaining best performance.

## H.3. Full $\beta$ Ablations

Figure 16 outlines the full suite of $\beta$ ablations for $\pi$-Distill variants and self-distillation for `Qwen3-4/8B`. We sweep over $\beta \in \{0.1, 0.25, 0.5\}$.

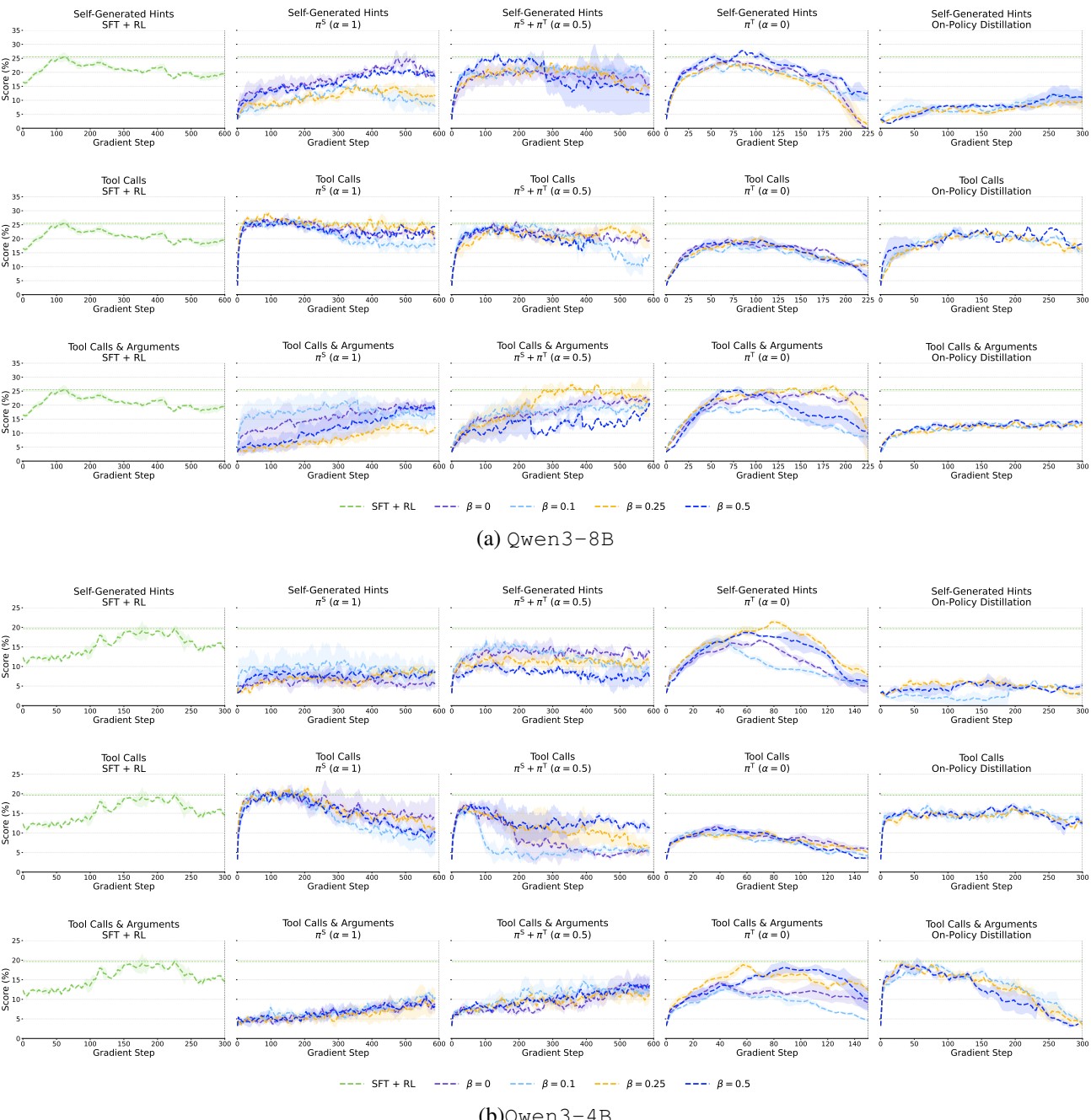

(a) `Qwen3-8B`

(b)`Qwen3-4B`

*Figure 16.* Full sweep for $\beta$ over variants of $\pi$-Distill. All experiments are given a 600 gradient step budget, where here we cut off experiment types early if they crash or do not continue learning. Consistent performance gains are seen for $\beta > 0$ across both 8B and 4B scales. We find that $\beta > 0$ is important in 14/18 cases. Specifically, we find it most important when training $\pi^T$ and less important with student-only training $\pi^S$.

# I. Implementation setup - Further details

## I.1. KL Estimator

For all losses requiring a Kl estimation, we use the Rae-Blackwellized estimator(Amini et al., 2025). For $\pi$-Distill, we use a sequence-level penalty aggregated into the reward term following Shah et al. (2026). In this setting, we allow the KL penalty to be absorbed into the advantage computation.

For OPSD, we use the same estimator as in (Amini et al., 2025) and directly back-propagate through the estimation.

## I.2. Length Penalty

In our setting, we found that policies can become overly verbose. To mitigate this, we add a cosine-shaped length penalty inspired by the cosine length-scaling reward (Yeo et al., 2025). We apply this penalty only to successful traces (i.e., when the base reward $r > 0$). For each assistant turn $i$ with token length $l_i$, we use a no-penalty threshold $l_{th} = 2000$ and a soft allowance $l_{max} = 5000$. If $l_i \leq l_{th}$, the turn penalty is $p_i = 0$ otherwise we assign a negative penalty using a linear–cosine schedule that increases in magnitude with length, approaching $-\lambda$ near $l_{max}$ with $\lambda = 0.1$, and becoming harsher beyond $l_{max}$ (up to an endpoint of $-2\lambda$). We then average the per-turn penalties across the $N$ assistant turns:

$$\bar{p} = \frac{1}{N} \sum_{i=1}^{N} p_i.$$

We cap the total penalty as $p_{total} = \max(\bar{p}, -0.3)$ and add that to the base reward $r' = r + p_{total}$.

## I.3. Hyper-Parameters

Here Table 2 we outline the swept hyperparameters, as well as their final values.

| Category | Parameter | R1-Distill-Llama-8B | QWEN3-4B | QWEN3-8B |
|---|---|---|---|---|
| 4*General | Seeds | 3 | 3 | 3 |
| | Rollout temperature | 0.75 | 0.75 | 0.75 |
| | Trace length filter | Discard if tokens $> 25$k ($\tau$-Bench/TP), $> 35$k (RL/OPSD) | | |
| | Advantage processing | pop zero-advantage (always) | | |
| 2*Training Budgets | $\tau$-Bench total gradient steps | 600 | 600 | 600 |
| | TP total gradient steps | 400 | 400 | 400 |
| 3*Sampling | Gradient steps per sampling | $\tau$-Bench = 3, TP = 2 | | |
| | Repeats per group | $\tau$-Bench = 5, TP = 4 | | |
| | Training tasks sampled | $\tau$-Bench = 64, TP = 45, SFT+RL ($\tau$-Bench) = 128 | | |
| 4*Optimization | Learning-rate sweep | {1e-6, 5e-6, 1e-5} | | |
| | Final LR | $\tau$-Bench: 5e-6 for all, TP: $\pi$-distill = 1e-5, RL/OPSD = 5e-6 | | |
| | $\beta$ | TP/OPSD = 0.5, $\pi$-distill = 0.25 (unless swept) | | |
| | Clipping/epsilon | Lower = 0.8, Upper = 1.2 | | |
| Alpha Annealing | $\alpha = 0.5$ schedule | Linearly anneal $\alpha : 0 \rightarrow 0.5$ over 15 epochs | | |

*Table 2.* Hyperparameters used for each fine-tuned model. All models share identical hyperparameter settings. Travel Planner is abbreviated to TP.

# J. Plot and Table Discrepancies

In this Fig 17, the 31.11% number is computed from the three seed peaks in the right panel. Each seed reaches its best score at a different gradient step, roughly one around 70 steps, another around 95, and another around 150. We take those three peak values and average them, which gives 31.11%, and that is what is reported in the table. The left panel instead shows the mean score at each gradient step, averaging the seeds at the same step, so its peak corresponds to the best point of the averaged curve rather than the average of the three best points. This difference in aggregation means the table can be higher than the peak of the averaged curve shown in the left plot.

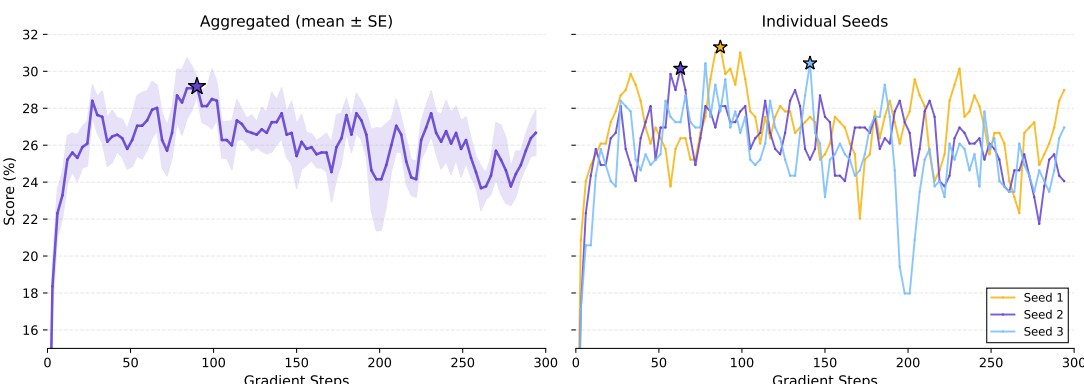

*Figure 17.* Comparison aggregating training runs versus the individual training runs. The left figure displays the mean score across seeds at each gradient step, while the right panel highlights the individual trajectories where peaks occur at different intervals. The reported table value (31.11%) represents the average of these individual seed peaks.

## K. Reward Hacking in Travel Planner

Our early experiments show that the model is able to consistently reward-hack the original rubric-based rewards proposed by (Xie et al., 2024). Specifically, Figure 18 shows the learning curves under the original reward structure. The policy appears to steadily improve; however, upon inspection, we observe that all traces collapse to the response structure shown in Section K. We attribute this behavior to the model learning that invoking the `Planner[.]` tool ends the conversation and, with certain arguments, reliably yields a high reward.

We address this issue by removing the requirement that all "easy" constraints must be satisfied before any "hard" constraint is checked. Instead, we evaluate each easy constraint individually and, only if it is satisfied, we then check its corresponding hard constraint.

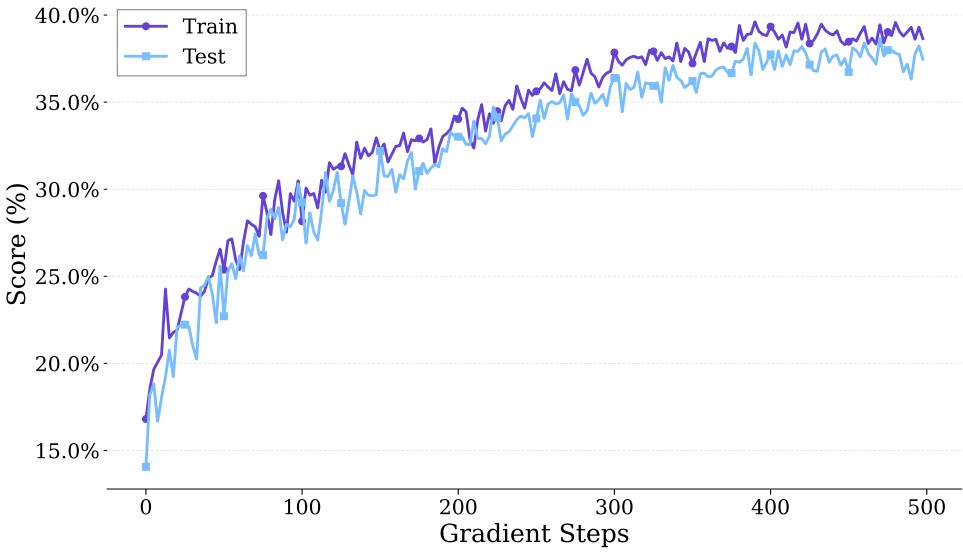

*Figure 18.* Training and testing curves for Travel Planner, using the original reward structure proposed in (Xie et al., 2024) with `Qwen3-4B`. We find that while the model learns, the sequences generally converge on reward hacking. Where the entire score is dominated by the "easy constraint score" as the model has learnt to satisfy some basic constraints, leading to the policy outputting the same response for all queries.

---

**τ-Bench Travel Planner Model Trajectory**

**Role:** Assistant

```
<think>
Okay, let's tackle this travel planning query. The user wants a 7-day plan from Punta
    Gorda to three Wisconsin cities between March 16-22, 2022, with a $5,700 budget.

First, I need to use the Planner tool to structure the itinerary. Check FlightSearch
    for available flights from Punta Gorda to Wisconsin cities. Maybe Milwaukee,
    Madison, and Green Bay? Use AccommodationSearch for each city. RestaurantSearch
    and AttractionSearch to suggest activities. Check GoogleDistanceMatrix for travel
    times between cities. Make sure the total costs are under budget. Use
    NotebookWrite to document the plan. Ensure the Planner incorporates all elements
    and stays within the budget. Avoid mentioning hints, just present the plan
    logically.
</think>

<action>Planner[Query: Craft a 7-day Wisconsin itinerary from Punta Gorda, including
    3 cities, flights, accommodations, attractions, and budget adherence using
    available tools. Prioritize March 16-22, 2022.]</action>
```

---

## L. Prompts

### L.1. System Prompts

For both τ-Bench and Travel Planner, we slightly modify the prompt to indicate that the model should output its action in corresponding `<action></action>` tags. Below, we display the full system prompts used for τ-Bench and Travel Planner.

---

**τ-Bench Retail System Prompt**

# Retail agent policy

As a retail agent, you can help users cancel or modify pending orders, return or exchange delivered orders, modify their default user address, or provide information about their own profile, orders, and related products.

- At the beginning of the conversation, you have to authenticate the user identity by locating their user id via email, or via name + zip code. This has to be done even when the user already provides the user id.

- Once the user has been authenticated, you can provide the user with information about the order, product, profile information, e.g. help the user look up the order id.

- You can only help one user per conversation (but you can handle multiple requests from the same user), and must deny any requests for tasks related to any other user.

- Before taking consequential actions that update the database (cancel, modify, return, exchange), you have to list the action detail and obtain explicit user confirmation (yes) to proceed.

- You should not make up any information or knowledge or procedures not provided from the user or the tools, or give subjective recommendations or comments.

- You should at most make one tool call at a time, and if you take a tool call, you should not respond to the user at the same time. If you respond to the user, you should not make a tool call.

## Domain basic

- All times in the database are EST and 24 hour based. For example "02:30:00" means 2:30 AM EST.

- Each user has a profile of its email, default address, user id, and payment methods. Each payment method is either a gift card, a paypal account, or a credit card.

---

- Our retail store has 50 types of products. For each type of product, there are variant items of different options. For example, for a 't shirt' product, there could be an item with option 'color blue size M', and another item with option 'color red size L'.

- Each product has an unique product id, and each item has an unique item id. They have no relations and should not be confused.

- Each order can be in status 'pending', 'processed', 'delivered', or 'cancelled'. Generally, you can only take action on pending or delivered orders.

- Exchange or modify order tools can only be called once. Be sure that all items to be changed are collected into a list before making the tool call!!!

## Cancel pending order

- An order can only be cancelled if its status is 'pending', and you should check its status before taking the action.

- The user needs to confirm the order id and the reason (either 'no longer needed' or 'ordered by mistake') for cancellation.

- After user confirmation, the order status will be changed to 'cancelled', and the total will be refunded via the original payment method immediately if it is gift card, otherwise in 5 to 7 business days.

## Modify pending order

- An order can only be modified if its status is 'pending', and you should check its status before taking the action.

- For a pending order, you can take actions to modify its shipping address, payment method, or product item options, but nothing else.

### Modify payment

- The user can only choose a single payment method different from the original payment method.

- If the user wants the modify the payment method to gift card, it must have enough balance to cover the total amount.

- After user confirmation, the order status will be kept 'pending'. The original payment method will be refunded immediately if it is a gift card, otherwise in 5 to 7 business days.

### Modify items

- This action can only be called once, and will change the order status to 'pending (items modifed)', and the agent will not be able to modify or cancel the order anymore. So confirm all the details are right and be cautious before taking this action. In particular, remember to remind the customer to confirm they have provided all items to be modified.

- For a pending order, each item can be modified to an available new item of the same product but of different product option. There cannot be any change of product types, e.g. modify shirt to shoe.

- The user must provide a payment method to pay or receive refund of the price difference. If the user provides a gift card, it must have enough balance to cover the price difference.

## Return delivered order

- An order can only be returned if its status is 'delivered', and you should check its status before taking the action.

- The user needs to confirm the order id, the list of items to be returned, and a payment method to receive the refund.

- The refund must either go to the original payment method, or an existing gift card.

- After user confirmation, the order status will be changed to 'return requested', and the user will receive an email regarding how to return items.

## Exchange delivered order

- An order can only be exchanged if its status is 'delivered', and you should check its status before taking the action. In particular, remember to remind the customer to confirm they have provided all items to be exchanged.

- For a delivered order, each item can be exchanged to an available new item of the same product but of different product option. There cannot be any change of product types, e.g. modify shirt to shoe.

- The user must provide a payment method to pay or receive refund of the price difference. If the user provides a gift card, it must have enough balance to cover the price difference.

- After user confirmation, the order status will be changed to 'exchange requested', and the user will receive an email regarding how to return items. There is no need to place a new order.

## Available tools:

```
{tools_info}
```

---

## Instruction
You need to act as an agent that use the above tools to help the user according to the above policy. At each step, your generation should have exactly the following format:

```
<think>
...Few lines of reasoning
</think>

<action>
{"name": <The name of the action>, "arguments": <The arguments to the action in json
    format>}
</action>
```

The Action will be parsed, so it must be valid JSON and within the `<action>` and `</action>` tags. You should not use made-up or placeholder arguments.
For example, if the user says "I want to know the current weather of San Francisco", and there is such a tool available:

```
{
    "type": "function",
    "function": {
        "name": "get_current_weather",
        "description": "Get the current weather",
        "parameters": {
            "type": "object",
            "properties": {
                "location": {
                    "type": "string",
                    "description": "The city and state, e.g. San Francisco, CA",
                },
                "format": {
                    "type": "string",
                    "enum": ["celsius", "fahrenheit"],
                    "description": "The temperature unit to use. Infer from location.",
                },
            },
            "required": ["location", "format"],
        },
```

```
    }
}
```

## Example response
**Step 1:**

```
<think>
... Few lines of reasoning
</think>

<action>
{"name": "get_current_weather", "arguments": {"location": "San Francisco, CA", "format
    ": "fahrenheit"}}
</action>
```

The tool and the user have the same id tags so if the user returns "70F", your response can be:
**Step 2:**

```
<think>
... Few lines of reasoning
</think>

<action>
{"name": RESPOND_ACTION_NAME, "arguments": {"RESPOND_ACTION_FIELD_NAME": "The current
    weather of San Francisco is 70F."}}
</action>
```

## Requirement
Try to be helpful and always follow the policy. Always try to validate your steps in your thinking and checkover your work, try to predict what will happen given your actions. Always make sure you generate valid JSON only.

## Important Notes

- Be very brief in the reasoning, do not repeat the entire context or the tools, just focus on what you need to do next.

- Always respond to the user using the tool {RESPOND_ACTION_NAME} to ensure the user sees your response.

- Only use a single set of thinking tags, the user cannot see your thoughts.

- Always wrap your tool calls `<action>` and `</action>` tags or else the system will not be able to parse your actions.

- Always start your outputs by thinking using `<think> ... </think>`.

Ok with this said, let us reason this out step by step always starting with `<think>` and ending with `</think>`.

---

**Travel Planning System Prompt**

You are a travel planning assistant. You have access to the following tools:

- `FlightSearch[Departure City, Destination City, Date]`

- `AccommodationSearch[City]`

- `RestaurantSearch[City]`

- `AttractionSearch[City]`

- `CitySearch[State]`

- `GoogleDistanceMatrix[Origin, Destination, Mode]`

- `NotebookWrite[Description]`

- `Planner[Query]`

Respond in this format:

```
<think>Your reasoning here</think>
<action>ToolName[arguments]</action>
```

Query:

```
{query}
```

## L.2. Full trajectories

**Example Trajectory**

**User Task:**

I'd like to start a return for #W6619432 and process it through PayPal account paypal_3738584 for the Dumbbell Set and Yoga Mat.

**Agent Trajectory:**

1. **Turn 1:**

```
<action>
{"name": "respond", "arguments": {"content": "I'd be happy to help you with your
    return for order #W6619432. First, I need to verify your identity to ensure the
     security of your account. Could you please provide your email address?
    Alternatively, you can provide your first name, last name, and zip code."}}
</action>
```

2. **Turn 2:**

```
<action>
{"name": "find_user_id_by_name_zip", "arguments": {"first_name": "Sophia", "
    last_name": "Nguyen", "zip": "20171"}}
</action>
```

3. **Turn 3:**

```
<action>
{"name": "get_order_details", "arguments": {"order_id": "#W6619432"}}
</action>
```

4. **Turn 4:**

```
<action>
{"name": "respond", "arguments": {"content": "Thank you for providing your
    information. I've verified your order #W6619432, which is delivered and
    includes:\n- Yoga Mat (item ID: 6195938807)\n- Dumbbell Set (item ID:
    3735133539)\n\nYou're requesting to return both items and process the refund
    through your PayPal account (paypal_3738584). The total refund amount will be
    $612.35.\n\nPlease confirm that you want to proceed with returning both items
    and receiving the refund to your PayPal account. Type 'yes' to confirm or 'no'
    to cancel."}}
</action>
```

5. **Turn 5:**

```
<action>
{"name": "return_delivered_order_items", "arguments": {"order_id": "#W6619432", "
    item_ids": ["6195938807", "3735133539"], "payment_method_id": "paypal_3738584
    "}}
</action>
```

6. **Turn 6:**

```
<action>
{"name": "respond", "arguments": {"content": "Your return request for order #
    W6619432 has been processed successfully. The order status is now 'return
    requested'. You will receive an email with detailed instructions on how and
    where to return the Yoga Mat and Dumbbell Set. The refund of $612.35 will be
    processed to your PayPal account (paypal_3738584). Please check your email for
    the return instructions. Is there anything else I can help you with today?"}}
</action>
```

7. **Turn 7:**

```
<action>
{"name": "respond", "arguments": {"content": "You're very welcome! I'm glad I could
     help with your return. Your refund will be processed to your PayPal account.
    Have a great day!"}}
</action>
```

## L.3. Self-generated hints

**Self-Generated Hint Example**

**Task:**
Your name is Amelia Wilson and your email is `amelia.wilson1598@example.com`. You are confident, cautious, dependent, shy, pessimistic. Cancel order #W3062096 because ordered by mistake.

**Self-Generated Hint:**
The agent authenticated the user via email, retrieved the order details, confirmed the order ID and cancellation reason, and successfully cancelled the order using the `cancel_pending_order` function. The refund process was initiated, and the user was informed of the next steps.

**Conversation:**

```
[user] Hi, I need to cancel an order I placed by mistake.

[assistant] Action: {"name": "respond", "arguments": {"content":
"I can help you cancel your order. First, I need to authenticate
your identity. Could you please provide your email address?
Alternatively, you can provide your first name, last name, and
zip code."}}

[user] Sure, my email is amelia.wilson1598@example.com.

[assistant] Action: {"name": "find_user_id_by_email",
"arguments": {"email": "amelia.wilson1598@example.com"}}

[API output] amelia_wilson_4614

[assistant] Action: {"name": "respond", "arguments": {"content":
"Thank you for providing your email. Could you please share the
```

order ID you'd like to cancel? This will help me verify the order
details and proceed with the cancellation."}}

[user] Yes, the order ID is W3062096.

[assistant] Action: {"name": "get_order_details",
"arguments": {"order_id": "#W3062096"}}

[API output] {"order_id": "#W3062096",
"user_id": "amelia_wilson_4614", "address": {"address1":
"388 Elm Avenue", "address2": "Suite 384", "city": "Dallas",
"country": "USA", "state": "TX", "zip": "75215"}, "items":
[{"name": "T-Shirt", "product_id": "9523456873",
"item_id": "3234800602", "price": 46.66, "options":
{"color": "red", "size": "L", "material": "cotton",
"style": "v-neck"}}, {"name": "Smart Watch",
"product_id": "6945232052", "item_id": "8739626972",
"price": 370.87, "options": {"color": "silver",
"band material": "silicone", "display": "AMOLED"}},
{"name": "Jigsaw Puzzle", "product_id": "1808611083",
"item_id": "9370300555", "price": 45.9, "options":
{"pieces": "1000", "theme": "art", "difficulty level":
"expert"}}], "fulfillments": [], "status": "pending",
"payment_history": [{"transaction_type": "payment",
"amount": 463.43, "payment_method_id": "paypal_4101143"}]}

[assistant] Action: {"name": "respond", "arguments": {"content":
"I can cancel your order #W3062096, which includes a T-Shirt,
Smart Watch, and Jigsaw Puzzle, totaling $463.43. The reason for
cancellation is 'ordered by mistake'. The refund will be processed
back to your PayPal account in 5-7 business days. Please confirm
with 'yes' if you'd like to proceed with the cancellation."}}

[user] Yes, please go ahead and cancel it.

[assistant] Action: {"name": "cancel_pending_order",
"arguments": {"order_id": "#W3062096",
"reason": "ordered by mistake"}}

[API output] {"order_id": "#W3062096",
"user_id": "amelia_wilson_4614", "address": {"address1":
"388 Elm Avenue", "address2": "Suite 384", "city": "Dallas",
"country": "USA", "state": "TX", "zip": "75215"}, "items":
[{"name": "T-Shirt", "product_id": "9523456873",
"item_id": "3234800602", "price": 46.66, "options":
{"color": "red", "size": "L", "material": "cotton",
"style": "v-neck"}}, {"name": "Smart Watch",
"product_id": "6945232052", "item_id": "8739626972",
"price": 370.87, "options": {"color": "silver",
"band material": "silicone", "display": "AMOLED"}},
{"name": "Jigsaw Puzzle", "product_id": "1808611083",
"item_id": "9370300555", "price": 45.9, "options":
{"pieces": "1000", "theme": "art", "difficulty level":
"expert"}}], "fulfillments": [], "status": "cancelled",
"payment_history": [{"transaction_type": "payment",
"amount": 463.43, "payment_method_id": "paypal_4101143"},
{"transaction_type": "refund", "amount": 463.43,
"payment_method_id": "paypal_4101143"}],
"cancel_reason": "ordered by mistake"}

[assistant] Action: {"name": "respond", "arguments": {"content":
"Your order #W3062096 has been successfully cancelled. The refund
of $463.43 will be processed back to your PayPal account within

```
5-7 business days. If you have any further questions or need
assistance with anything else, feel free to ask!"}}

[user] Thank you, that's all I needed. ###STOP###
```

**Hint Generation Prompt:**
You are creating a concise natural language hint that will help an agent solve the task. Write a helpful hint describing the key actions and strategy the agent used. If the agent did not fully complete the task, focus on what they did achieve. IMPORTANT: Output ONLY plain text - no JSON, no code blocks, no markdown formatting. Keep the hint to 2-3 sentences maximum.

**The prompt used**

### L.4. Privileged Information Prompts

We use the same prompts When using PI in both models. Specifically, we try to ask the model not to reveal it has access to PI by encasing it in `<Secret information></Secret information>` tags. We found this to be unsuccessful as shown in the experiments in App H.

**Privileged Instructions**

`<Secret information>`
This section contains privileged hints derived from prior successful trajectories. Use them to stay on track, but never mention that hints were provided.
**Hint Summary:**

`{hint_lines}`

**Your Task:**

1. Follow the hints to accomplish the task still reasoning with `<think>...</think>` before actions.

2. Never acknowledge having access to secret hints.

Always start responses with a `<think>` block.
`</Secret information>`

**Privileged Tool Instructions**

`<Secret information>`
This section is highly important for you to pay attention to. You have been given access to secret information that is not available to the user and should not be mentioned.
Here is a successful set of tools, given the context and tools that have been called so far use this information to accomplish the task.

`{tool_calls_section}`

**Your Task:**

1. **You will need to use the provided tools in order to accomplish the task.**

2. **Reasoning:** Please reason about your steps. If you think you need to use a tool, provide reasoning justifications and clear logic for using it before invoking it using `<think>reasoning...</think>`.

3. **Strict Constraint:** Do NOT mention that you have been given access to the secret information. You will be penalized for violating this rule.

**Reasoning Format**

Your reasoning traces will be used to train future agents that do not have access to privileged information. Please make sure your reasoning is clear and concise.
Always start your outputs by thinking using `<think> ...</think>`
`</Secret information>`

