# OpenReview forum: "Privileged Information Distillation for Language Models"
_ICML.cc/2026/Conference — ICML 2026 regular_

### Official Review · Reviewer_e8B1 · 2026-03-04

**Soundness:** 3
**Presentation:** 2
**Significance:** 2
**Originality:** 3
**Overall Recommendation:** 4
**Confidence:** 2

**Summary:**

This paper presents π-Distill, a joint teacher–student RL framework that distills agentic capabilities from closed-source models using privileged information (PI) when CoT is unavailable. By co-training a PI-informed teacher and a PI-blind student via shared parameters, the method outperforms standard SFT with CoT + RL baselines and demonstrates robust OOD generalization.

**Compliance With Llm Reviewing Policy:**

Affirmed.

**Key Questions For Authors:**

Q1: Teacher Performance & Transfer Efficiency: Could the authors provide the performance of the privileged teacher model to quantify the framework's information transfer efficiency from teacher to student?
Q2: PI Accessibility & Robustness: Could the authors discuss the framework's sensitivity to scarce, noisy, or costly PI, and how distillation performance degrades if the privileged cues are imperfect?
Q3: Computational Efficiency: Could the authors provide the actual computational costs (e.g., GPU hours, training time) of π-Distill?

**Limitations:**

This paper lacks a section or discussion addressing the limitations of the proposed framework.

**Strengths And Weaknesses:**

Strength:
- Addressing Black-box Constraints: The paper focuses on a realistic and challenging scenario where the teacher is a black-box model, providing a solution to distill agentic capabilities without accessing CoT information.
- Generalization & Robustness: The method demonstrates strong task-agnostic generalizability, particularly excelling in OOD (Out-of-Distribution) scenarios, proving that the student model acquires fundamental agentic capabilities.
- Comprehensive Evaluation: The paper presents a rigorous and well-structured experimental design, supported by various agentic tasks. The inclusion of OOD benchmark and detailed ablation studies provides solid empirical evidence for the framework's effectiveness.

Weakness:
- Outdated References: Some citations remain in preprint (arXiv) format despite being formally published. For example, "TravelPlanner: A Benchmark for Real-World Planning with Language Agents" should be updated to its official version in PMLR 2020.
- Structural Imbalance: The current organization of the paper appears somewhat imbalanced. Specifically, Section 3 is notably brief compared to other sections, which makes the overall structure feel fragmented rather than a cohesive progression. Integrating this content into Section 2 or 4 as a subsection would likely enhance the structural cohesion and ensure a more seamless progression for the reader.
- Missing Literature Context: While the experiments thoroughly compare RL, SFT, and CoT-based methods, the Related Work (Section 9) lacks a dedicated discussion of these foundational paradigms.
- Missing Teacher Upper-bound: The experiments lack the performance of the teacher model with privileged information, making it impossible to quantify the information transfer efficiency of the RL framework.
- PI Accessibility: High-quality Privileged Information (PI) is often scarce or costly in real-world tasks, yet the framework assumes its constant availability during training. This creates a risk of mimicking suboptimal teacher behaviors when privileged cues are noisy or imperfect.

---

> ### Author Rebuttal · Authors · 2026-03-31
>
> We thank the reviewer for all the comments and these references, and are happy to discuss them.
>
> ---
>
> ### W1: Outdated references
>
> We thank the reviewer for pointing this out. We have updated these references to their published versions and this will be reflected in our next revision.
>
> ---
>
> ### W2: Structural imbalance (Section 3)
>
> We appreciate this structural feedback. Section 3 was intended to serve as a standalone motivation for why training-time PI is valuable before introducing the method. However, we agree that its brevity makes the paper feel fragmented. In our revision, we will integrate Section 3 as a subsection within Section 4 (Methods), creating a smoother progression from background to motivation to our proposed framework.
>
> ---
>
> ### W3: Missing literature on RL/SFT/CoT paradigms
>
> We thank the reviewer for this suggestion. In our revision, we have added a dedicated subsection to Related Work discussing these foundational paradigms:
>
> "**SFT and RL Post-Training for Agents.** The standard pipeline for training capable LM agents involves supervised fine-tuning on frontier model outputs followed by reinforcement learning. This paradigm has been adopted across scales, from large-scale pre-trained models like DeepSeek-V3 (DeepSeek-AI et al., 2025) which employ SFT and RL stages to unlock capabilities, to specialized agentic systems such as INTELLECT-3 (Senghaas et al., 2025) which scales multi-turn RL with tool use, and domain-specific agents like Chroma Context-1 (Bashir et al., 2026) which trains a search agent via SFT on frontier trajectories followed by GRPO-based RL. Our work builds on this pipeline but addresses the specific challenge where the frontier model's full reasoning traces are unavailable, a common constraint with closed-source systems."
>
> ---
>
> ### W4: Missing teacher upper-bound
>
> The teacher upper bound is the performance of DeepSeek V3.1 Chat-671B, which we report in Table 1: **45.0%** on Travel Planner, **51.3%** on τ-Bench Retail, and **40.0%** on τ-Bench Airline.
>
> Compared to this upper bound, our best π-Distill configurations achieve remarkable transfer efficiency. Most notably, **Qwen3-8B with π-Distill (α=1) reaches 44.1% on Travel Planner, nearly matching the 671B frontier model (45.0%)** with an 84x smaller model. On τ-Bench Retail, π-Distill (α=0) achieves 31.1% vs. DeepSeek's 51.3%, reaching ~61% of the teacher's performance.
>
> ---
>
> ### W5: PI accessibility (noisy / scarce / costly)
>
> We address the three facets of this concern:
>
> **Noisy/imperfect PI:** We study this directly in Section 7 and Figure 2 using three PI types of varying quality. When PI provides low initial utility (Δ ≈ 0 or Δ < 0), teacher-training variants of π-Distill (α > 0) can still extract value: as shown in Figure 11 (App E), the teacher *learns* to leverage even initially noisy PI, consistently achieving positive Δ_max. This demonstrates that π-Distill is designed to handle suboptimal PI through joint training.
>
> **Scarce PI:** π-Distill is naturally compatible with partially available PI. One can apply PI for tasks where it is available and fall back to standard RL for tasks where it is not. The joint objective handles this seamlessly since the teacher and student share parameters.
>
> **Costly PI:** Our algorithm requires only *one* PI annotation per training query (a single successful trajectory), which is significantly cheaper than the multiple full traces needed for SFT/SFT+RL pipelines. Thus π-Distill not only outperforms these baselines but is also more resource-efficient in terms of PI collection.
>
> ---
>
> ### Q1: Teacher performance & transfer efficiency
>
> We provide the teacher's relative performance on training tasks in Figure 11 (App E), where Δ_max measures how well each algorithm leverages PI compared to pure RL. For evaluation tasks, the teacher upper bound is DeepSeek V3.1 Chat-671B (Table 1). As noted above, π-Distill achieves remarkable transfer efficiency. For example, Qwen3-8B reaches 44.1% on Travel Planner, nearly matching DeepSeek's 45.0%.
>
> ---
>
> ### Q2: PI robustness
>
> Please see our detailed response to W5 above.
>
> ### Q3: Computational costs
> Our largest-scale setting is Qwen3-8B on τ-Bench (longest reasoning traces). Per epoch (4 steps) on 2 H100 GPUs, π-Distill (α = 1/0) takes 37.2 ± 4.21 GPU hours, joint training (α = 0.5) takes 39.0 ± 5.43, and vanilla RL takes 32.6 ± 3.97. Each experiment runs up to 600 steps, with early termination if the run collapses.
>
> We report the runtime for some representative runs:
>
> **α= 1:** 3d 23h 37m 27s
>
> **α= 0:** 3d 18h 23m 02s
>
> **α= 0.5:** 4d 13h 21m 14s
>
> **Raw RL + SFT:** 3d 02h 11m 20s +  21h 16m 34s
>
> When accountinf for SFT time, both RL + SFT (best baseline) and π-Distill are fairly comparable. We also find that π-Distill converges faster than RL. We will add full convergence and runtime comparison plots in our revision. We are happy to provide any further details.

---

> > ### Author Rebuttal · Reviewer_e8B1 · 2026-04-06
> >
> > My concerns have been adequately addressed.

---

> > > ### Author Response · Authors · 2026-04-06
> > >
> > > We thank the reviewer for their feedback and happy their concerns are resolved. We hope our rebuttal improved your confidence as well as perceived significance of our work. Given all your concerns are resolved we kindly suggest increasing your recommendation score.
> > >
> > >
> > > ---
> > >
> > > Further, we would like to take the added space to inform the reviewer of additional experiments we have conducted since submission that we did not have space to provide in our initial response.
> > >
> > > ---
> > >
> > > **Updated OPCD results.** We expanded OPCD to all three PI types (previously only Tool Calls). This improved OPCD's best-case performance on Travel Planner:
> > >
> > > | Model | OPCD Travel Planner (old) | OPCD Travel Planner (updated) |
> > > | :--- | :--- | :--- |
> > > | R1-Distill-Llama-8B | 5.01% ± 0.91 | 13.1% ± 1.09 |
> > > | Qwen3-4B | 24.3% ± 0.05 | 29.8% ± 1.14 |
> > > | Qwen3-8B | 27.4% ± 1.47 | 37.5% ± 1.53 |
> > >
> > > All other results are unchanged and our final conclusions remain. π-Distill still consistently outperforms OPCD across all settings. This does show that OPCD is rather sensitive to PI but with the correct type can be a useful algorithm.
> > >
> > > ---
> > >
> > > **Extended PI analysis to OPCD (new Figure X).**  We extended the analysis from Section 7 to OPCD, finding that it exhibits different dynamics than π-Distill. For OPCD, higher KL is not always detrimental and the information richness of PI matters most, with Tool Calls & Arguments often performing best. However, excessive KL can override positive utility (e.g., Qwen3-8B on τ-Bench). These findings further highlight the advantage of π-Distill's joint training, which is more robust to high-KL regimes.
> > >
> > > ---
> > >
> > > **Extended β ablations.** We extended β ablations to Qwen3-4B and OPCD across both Qwen models on τ-Bench. Results are consistent with Qwen3-8B. β > 0 achieves top performance in 17/21 configurations. OPCD is not sensitive to the specific value of β (β ∈ {0.1, 0.25, 0.5}).
> > >
> > >
> > > ---
> > >
> > >
> > > We hope the reviewer finds these added experiments insightful and again thank them for taking the time to help us improve our work.

---

### Official Review · Reviewer_uiqL · 2026-03-04

**Soundness:** 3
**Presentation:** 4
**Significance:** 3
**Originality:** 3
**Overall Recommendation:** 5
**Confidence:** 3

**Summary:**

This paper studies how to exploit training-time privileged information (PI) to improve LM agents while still producing a test-time policy that does not require PI. It proposes π-Distill, a joint teacher-student training scheme implemented via a single shared-parameter model, where the teacher conditions on PI and the student does not, and both are optimized simultaneously with KL regularization. Results show that π-Distill generally matches or outperforms SFT+RL pipelines that assume access to CoT.

**Compliance With Llm Reviewing Policy:**

Affirmed.

**Final Justification:**

The authors addressed my concerns during the rebuttal phase; therefore, I raised my score to "Accept".

**Key Questions For Authors:**

1. In Table 1, why are the comparisons for R1-Distill-Llama and Qwen3 slightly different (i.e. no *RL* and *SFT w/o CoT + RL* for R1-Distill-Llama, and no *SFT w/ CoT + RL* for Qwen3)? Could you please complete the experimental results for each?

2. When you say *w/o CoT*, are reasoning tokens disabled/removed from rollouts, or is <think> always produced but hidden? Concretely, what token subsets are included in the loss/reward and in the distilled targets?

3. How sensitive are the Figure 2 conclusions to the estimator?

**Limitations:**

yes

**Strengths And Weaknesses:**

**Strengths:**

**S1. Clear motivation:** The motivation of this paper is clear and reasonable.

**S2. Extensive experiments and analysis:** Empirical results are fairly broad: different benchmarks, and across three base models. The analysis in Section 7 is thorough and insightful.

**S3. Excellent writing and presentation:** The paper is clearly written and logically structured; the charts and graphs are very helpful.

**Weaknesses:**

**W1. Limited comparison:** The baselines are relatively simple and lacks comparisons with related works.

**W2. Lack of quantitative predictive evaluation:** Figure 2 is suggestive, but there is no explicit predictive test (e.g., correlation coefficients, or held-out prediction).

---

> ### Author Rebuttal · Authors · 2026-03-31
>
> We thank the reviewer for all the comments and these references, and are happy to discuss them.
>
> ---
>
> ### W1: Limited baselines
>
> We appreciate this feedback and want to clarify the breadth of our comparisons. Our baselines cover the key paradigms in this space:
>
> 1. **Standard RL and SFT baselines** (RL, SFT w/ CoT, SFT w/o CoT, SFT + RL): The industry-standard post-training pipeline.
> 2. **On-Policy Context/Self Distillation (OPCD)**: We concurrently and independently propose this method (Sec4 and App A.2), which is equivalent to the algorithms in [Zhao et al., 2026; Shenfeld et al., 2026; Qu et al., 2026] (self-distillation). Beyond proposing OPCD, we introduce π-Distill and show it consistently outperforms OPCD.
> 3. **Variational EM**: Compared in App A/B (Figure 5), showing π-Distill's superiority.
> 4. Concurrent PI methods (iGRPO, RLTF, Experiential RL [Hatamizadeh et al.; Song et al.; Shi et al., 2026]): These are subsumed by our α=1, β=0 ablation. We show that adding KL regularization (β > 0) and joint
>   training consistently improves over this configuration (7/8 settings, Sec 8 + App C.3). Additionally, each uses only a single PI source, whereas we explore multiple and characterize when each succeeds or fails
>   (Sec 7).  .
>
> Furthermore, the industry standard in agentic settings is SFT w/ CoT + RL, which we match or beat despite not having access to CoT traces. We are happy to consider additional baselines if the reviewer has specific methods in mind.
>
> ---
>
> ### W2: No quantitative predictive evaluation for Figure 2
>
>   We agree that Figure 2 is suggestive rather than predictive, and will note this as an explicit limitation. A fully controlled quantitative study would need to isolate KL, hint utility, and model capacity, which Figure 2 varies simultaneously. That said, to our knowledge, our work is the only one, including concurrent works [1-5] and other PI methods (iGRPO, RLTF, Experiential RL), that systematically varies the source of PI and quantitatively studies its effects on learning. This analysis reveals that different PI types induce different KL divergences and initial utilities, which determine whether student-only, teacher-only, or joint training succeeds. Crucially, we not only identify these failure modes but provide a concrete solution: joint training (α = 0.5) robustly mitigates them, offering actionable guidance absent from concurrent work.
>
> ---
>
> ### Q1: Different baselines for R1-Distill-Llama vs Qwen3
>
> We clarify both directions:
>
> **R1-Distill-Llama-8B (no pure RL/SFT w/o CoT + RL):** As discussed in lines 264–267, R1-Distill-Llama-8B achieves 0% on both benchmarks without SFT warm-starting (see Base row in Table 1), failing to generate correct trajectories even when conditioned on PI. This makes direct RL training infeasible. We therefore warm-start it with SFT w/ CoT from expert traces. This setup is informative because it demonstrates that π-Distill remains beneficial *even when the model has already been SFT'd on traces containing the PI*.
>
> **Qwen3 (SFT w/ CoT + RL):** We do report SFT w/ CoT + RL for Qwen3. It appears as the last row in each Qwen section of Table 1, serving as our soft-upper bound baseline, as our method do not require/assume full access to the teacher's CoT. Which we show we can outeperform.
>
> ---
>
> ### Q2: What tokens are included w/o CoT?
>
>
> Thank you for this important clarification question. The token handling differs by stage:
>
> - **SFT w/ CoT:** All tokens from the frontier model's trajectory (reasoning + tool calls + arguments) are included in the SFT loss.
> - **SFT w/o CoT:** Only the tool call tokens are included in the SFT loss. Reasoning tokens are removed from the training data entirely.
> - **RL stage (all methods):** The model freely generates all tokens (including any reasoning it produces). All generated tokens are included in the loss and reward computation. We do not suppress or hide reasoning during RL.
> - **π-Distill PI targets:** The PI is placed in the teacher's context. The teacher generates its own trajectories conditioned on PI, and the student learns from these via importance-weighted RL.
>
> ---
>
> ### Q3: Sensitivity to KL estimator
>
> For Figure 7, we use the Rao-Blackwellized KL estimator (Amini et al., 2025), which to the best of our knowledge is unbiased and the lowest-variance existing estimator for KL divergence between language models. We compute our estimates using 5,000 samples per query, yielding very small error bars. Because the estimator is unbiased and low-variance, even an alternative unbiased estimator with higher variance should produce similar conclusions. Because of this, we believe our results provide accurate takeaways.

---

> > ### Author Rebuttal · Reviewer_uiqL · 2026-04-01
> >
> > Thank you very much for your response. I will consider increasing my score and confidence accordingly.

---

> > > ### Author Response · Authors · 2026-04-04
> > >
> > > We thank the reviewer for their feedback and happy their concerns are resolved. We would like to take the added space to inform the reviewer of additional experiments we have conducted since submission that we did not have space to provide in our initial response.
> > >
> > > ---
> > >
> > > **Updated OPCD results.** We expanded OPCD to all three PI types (previously only Tool Calls). This improved OPCD's best-case performance on Travel Planner:
> > >
> > > | Model | OPCD Travel Planner (old) | OPCD Travel Planner (updated) |
> > > | :--- | :--- | :--- |
> > > | R1-Distill-Llama-8B | 5.01% ± 0.91 | 13.1% ± 1.09 |
> > > | Qwen3-4B | 24.3% ± 0.05 | 29.8% ± 1.14 |
> > > | Qwen3-8B | 27.4% ± 1.47 | 37.5% ± 1.53 |
> > >
> > > All other results are unchanged and our final conclusions remain. π-Distill still consistently outperforms OPCD across all settings. This does show that OPCD is rather sensitive to PI but with the correct type can be a useful algorithm.
> > >
> > > ---
> > >
> > > **Extended PI analysis to OPCD (new Figure X).**  We extended the analysis from Section 7 to OPCD, finding that it exhibits different dynamics than π-Distill. For OPCD, higher KL is not always detrimental and the information richness of PI matters most, with Tool Calls & Arguments often performing best. However, excessive KL can override positive utility (e.g., Qwen3-8B on τ-Bench). These findings further highlight the advantage of π-Distill's joint training, which is more robust to high-KL regimes.
> > >
> > > ---
> > >
> > > **Extended β ablations.** We extended β ablations to Qwen3-4B and OPCD across both Qwen models on τ-Bench. Results are consistent with Qwen3-8B. β > 0 achieves top performance in 17/21 configurations. OPCD is not sensitive to the specific value of β (β ∈ {0.1, 0.25, 0.5}).
> > >
> > >
> > > ---
> > >
> > >
> > > We hope the reviewer finds these added experiments insightful and again thank them for taking the time to help us improve our work.

---

### Official Review · Reviewer_bR9d · 2026-03-13

**Soundness:** 3
**Presentation:** 3
**Significance:** 4
**Originality:** 3
**Overall Recommendation:** 5
**Confidence:** 3

**Summary:**

This paper introduces $\pi$-Distill, a framework to distill agentic capabilities from frontier language models into smaller models by treating frontier model outputs as privileged information. Specifically, the authors extract tool calls, tool calls with arguments, and summaries of trajectories of correct frontier model trajectories (where full traces are not available) to form privileged information. A teacher is formed with the same weights as the student policy, but with the privileged information in context. The authors then optimize a joint objective to simultaneously update both the teacher (maximize reward under a KL constraint with the student) and the student (maximize reward relative to an importance ratio with the teacher and under a KL constraint with the teacher).  These objectives are linearly mixed. The authors train their policy on tau-bench and travel-planner bench tasks and evaluate on a held-out set. They find that their approach is able to effectively transfer capabilities to the student and outperform SFT, RL, and on-policy context distillation baselines. Finally, the authors perform a suite of analyses to determine what type or format of privileged information is most effective, the effect on model size, and out-of-domain generalization, etc., as well as ablation of hyperparameters.

**Compliance With Llm Reviewing Policy:**

Affirmed.

**Final Justification:**

All my concerns were addressed - I have kept my accept rating.

**Key Questions For Authors:**

1. Instead of summarizing, did the authors try asking the student model to fill in the missing reasoning between tool calls and using this completed trajectory as privileged information and/or data for SFT? It seems like this might be an important baseline and could also improve the performance of the method, as the privileged information is potentially more detailed and easier to distill, as it matches the output format (reasoning and tool calls).

2. Does the performance of the teacher also increase during training? If so, by how much, and how much faster/slower is the increase compared to the student?

3. Do we really need to jointly train the student and the teacher together? Did the authors try a two-stage training process where the teacher is first updated with RL and then a second stage where only the student is updated?

**Limitations:**

yes

**Strengths And Weaknesses:**

### Strengths
1. The $\pi$-distill framework solves a novel and highly relevant problem, i.e., how to distill from frontier models when full model traces are not present. Additionally, the joint teacher-student training is novel as it enables the teacher to be a good reference while also improving its ability to use the privileged information.

2. The authors achieve strong empirical results, often outperforming standard SFT and RL baselines on both in-domain and out-of-distribution tasks.

3. The paper presents a host of well-executed analyses, such as understanding what type of privileged information is best to use and how to weight teacher training with student training, which will be useful for practitioners.

### Weaknesses
1. The related work section seems to be missing many of the recent papers on self-distillation / privileged information distillation that use a similarly initialized teacher that has access to privileged information but is initialized with the same weights [1-4]. It would be helpful for the authors to discuss the differences between these works and their own work, since it seems this line of work is most similar to their approach. Also, $\pi$-distill does not explicitly train the teacher to be a better teacher. The authors should compare their work to those that do this, like [5].

2. There seems to be a discrepancy in the 4B Qwen results. Specifically, the Qwen-4B trained model seems to outperform all baselines in-domain, but not out-of-domain. If it were the case that "SFT supervision aids smaller models" (line 326), why do the SFT and RL baselines perform as poorly in the Qwen-4B setting as the 8B model settings in domain?

3. While the presentation is generally quite good, Figure 2 is quite confusing as there are 2 sets of variables being varied along with 2 baselines, and the KL values. It would be helpful to split this figure into sub-figures or separate figures to make the point of the Figure easier to understand from quickly skimming it.

[1] https://arxiv.org/pdf/2601.20802

[2] https://arxiv.org/pdf/2601.18734

[3] https://arxiv.org/pdf/2601.19897

[4] https://arxiv.org/pdf/2602.02405

[5] https://arxiv.org/pdf/2601.18778

---

> ### Author Rebuttal · Authors · 2026-03-31
>
> We thank the reviewer for all the comments and these references, and are happy to discuss them.
>
>   [1-3]: We discuss these in App A.2, noting that we concurrently and independently propose the same core algorithm (On-Policy Context Distillation, OPCD, Sec 4/A.2). Crucially, our contribution goes well beyond
>   proposing OPCD. We introduce π-Distill, a strictly more general algorithm that consistently outperforms OPCD across all tested settings. We further provide significant additional ablations and studies.
>
>   [4]: Mendes et al. (2026) address a complementary challenge of transforming expert solutions into learnable formats via contrastive learning, without using a shared-parameter teacher-student framework, joint RL
>   training, or PI conditioning.
>
>   [5]: We want to clarify that π-Distill does explicitly train the teacher via J_Teacher (line 140), which optimizes the teacher to maximize reward while staying close to the student. The distinction from SOAR is
>   that it uses meta-RL where the teacher generates problems and is rewarded based on student improvement. This is distinct from π-Distill and both approaches can be used
>   complementarily. We will add a discussion of [5].
>
>  We also note that [1-5] were all made public less than two months before the ICML submission deadline, making them concurrent works.
>
>
>   ---
>   W2: 4B Qwen in-domain vs OOD discrepancy
>
>   While true, π-Distill only outperforms SFT w/ CoT + RL (the soft upper bound) in one in-domain setting for Qwen3-4B, so SFT w/ CoT + RL is still generally better for smaller models.
>
>   Out of domain, we believe the gap comes from smaller models learning more general skills during SFT that transfer better. Notably, this pattern does not hold for Qwen3-8B, where π-Distill achieves strong OOD gains
>    (12-14% on Airline vs 6-8% for baselines), suggesting that larger model capacity better supports generalization of RL-learned behaviors. We note these are hypotheses and future work should investigate this
>   further.
>
>   ---
>   W3: Figure 2 confusing
>
>   We agree that Figure 2 is information-dense, largely due to ICML's 9-page constraints requiring us to consolidate multiple analysis dimensions into a single figure. We will contemplate splitting the figures in future pre-print revisions.
>
>
>   ---
>   Q1: Infilling reasoning between tool calls?
>
>   Yes, we explored this in initial experiments (App B). We tried (1) infilling reasoning chains between tool calls using the student model and (2) training a model to perform infilling in a Quiet-STaR-like fashion.
>
> Neither yielded improvements. Even successful frontier model traces contain redundant or noisy steps, making coherent infilling difficult.
>
> We also tested a sequential Variational EM baseline (AppA/B), which outperformed these offline approaches but was still substantially worse than π-Distill. These negative results motivated our PI-based approach, which leverages the structure of successful traces rather than attempting to reconstruct their reasoning.
>
>   ---
>   Q2: Does teacher performance increase during training?
>
>   Yes, as analyzed in Figure 11 (App E). Figure 11 reports Δ_max, the difference between the maximum training score with PI versus without PI (pure RL). Configurations that train the teacher (α > 0) consistently
>   achieve positive Δ_max, indicating the teacher effectively learns to use PI to exceed pure RL performance. The teacher generally improves at least as fast as the student, and often faster due to the additional PI
>   signal. In our revision, we will add explicit training curves comparing teacher performance against pure RL.
>
>   ---
>   Q3: Two-stage training vs joint?
>
>
> See the sequential Variational EM baseline in Q1.

---

> > ### Author Rebuttal · Reviewer_bR9d · 2026-04-02
> >
> > Thank you for the response. I will keep my score and advocate for the acceptance of this work.

---

> > > ### Author Response · Authors · 2026-04-04
> > >
> > > We thank the reviewer for their feedback and happy their concerns are resolved. We would like to take the added space to inform the reviewer of additional experiments we have conducted since submission that we did not have space to provide in our initial response.
> > >
> > > ---
> > >
> > > **Updated OPCD results.** We expanded OPCD to all three PI types (previously only Tool Calls). This improved OPCD's best-case performance on Travel Planner:
> > >
> > > | Model | OPCD Travel Planner (old) | OPCD Travel Planner (updated) |
> > > | :--- | :--- | :--- |
> > > | R1-Distill-Llama-8B | 5.01% ± 0.91 | 13.1% ± 1.09 |
> > > | Qwen3-4B | 24.3% ± 0.05 | 29.8% ± 1.14 |
> > > | Qwen3-8B | 27.4% ± 1.47 | 37.5% ± 1.53 |
> > >
> > > All other results are unchanged and our final conclusions remain. π-Distill still consistently outperforms OPCD across all settings. This does show that OPCD is rather sensitive to PI but with the correct type can be a useful algorithm.
> > >
> > > ---
> > >
> > > **Extended PI analysis to OPCD (new Figure X).**  We extended the analysis from Section 7 to OPCD, finding that it exhibits different dynamics than π-Distill. For OPCD, higher KL is not always detrimental and the information richness of PI matters most, with Tool Calls & Arguments often performing best. However, excessive KL can override positive utility (e.g., Qwen3-8B on τ-Bench). These findings further highlight the advantage of π-Distill's joint training, which is more robust to high-KL regimes.
> > >
> > > ---
> > >
> > > **Extended β ablations.** We extended β ablations to Qwen3-4B and OPCD across both Qwen models on τ-Bench. Results are consistent with Qwen3-8B. β > 0 achieves top performance in 17/21 configurations. OPCD is not sensitive to the specific value of β (β ∈ {0.1, 0.25, 0.5}).
> > >
> > >
> > > ---
> > >
> > >
> > > We hope the reviewer finds these added experiments insightful and again thank them for taking the time to help us improve our work.

---

### Official Review · Reviewer_ELnF · 2026-03-17

**Soundness:** 4
**Presentation:** 3
**Significance:** 2
**Originality:** 3
**Overall Recommendation:** 4
**Confidence:** 4

**Summary:**

This paper proposes $\pi$-Distill which is teacher-student framework where the teacher has the access to the privilege information from another larger/more capable model, while the student and teacher both tries to optimize the similarity between the distribution and the reward. The results show that some variations of the propose method is the best among all tested models.

**Compliance With Llm Reviewing Policy:**

Affirmed.

**Key Questions For Authors:**

1. I think $J_{\text{Student}}$ is wrong, it should be $E_{o\sim\pi_{\theta}^T}$, from teacher's trajectories.
2. Related to 1, since the teacher has more privileged information, the policy distribution could be narrower than the student's policy, which makes the importance sampling less valid, i.e. the student's unsampled region is left unchanged.

**Limitations:**

yes

**Strengths And Weaknesses:**

# Strengthes

The method is theoretically sound and very intuitive. Although it should be able to generalize to any privilege information, not just that coming from a bigger model. The joint training idea also makes sense to me although it is expected that the performance is super sensitive to $\alpha$.

# Weaknesses

1. It seems that the improvement is not super significant if taking the std error into account.
2. I would expect this method to benefit from more multi-step benchmarks where the privilege information is more useful for the models to explore the correct method. But it was not tested on SWEBench/WebArena etc.
3. The values of $\alpha$ and $\beta$ requires careful tuning, which could limit the application of the method.

---

> ### Author Rebuttal · Authors · 2026-03-31
>
> We thank the reviewer for taking the time to aid us in imporiving our work.
>
>
> ### W1: Improvement significance
>
> We thank the reviewer for raising this point. We want to clarify that SFT w/ CoT + RL serves as a *soft upper bound*, as it uses strictly more information (full Chain-of-Thought traces) than our method. The key contribution of π-Distill is achieving competitive or superior performance *without* requiring CoT access, which is the realistic closed-source setting.
>
> Our main comparison points are the baselines that also lack CoT: SFT w/o CoT + RL and raw RL. We consistently observe **22–44% relative improvements** over the strongest CoT-free baselines. For instance, in τ-Bench Retail Qwen 8B:
>
>
> *Ratios relative to best baseline for Qwen3-8B: RL (23.9% ± 0.44)*
>
> | Method | Score | Ratio |
> | :--- | :--- | :--- |
> | On-Policy Context Distill | 25.8% ± 2.46 | **1.08x** |
> | π-Distill π (α = 0) | 31.1% ± 0.73 | **1.30x** |
> | π-Distill π + πh (α = 0.5) | 30.6% ± 0.67 | **1.28x** |
> | π-Distill πh (α = 1) | 29.7% ± 0.33 | **1.24x** |
>
> These gains are statistically significant. For Qwen3-8B, π-Distill variants have small standard errors (0.33–0.73) while the gaps over baselines are large. Furthermore, for Qwen3-8B, π-Distill (α=0) at 31.1% actually **surpasses** the soft upper bound SFT w/ CoT + RL (29.1% ± 2.14), demonstrating that our method can close and sometimes exceed the CoT soft-upperbound. Further these gains are similar for R1-distilled-Lamma and Qwen 4b.
>
> ---
>
> ### W2: More multi-step benchmarks (SWEBench/WebArena)
>
> We agree that extending to benchmarks like SWE-Bench and WebArena is a valuable direction. We want to emphasize that all of our benchmarks *are* multi-step agentic tasks: τ-Bench involves multi-turn customer service interactions requiring sequential tool calls (e.g., looking up orders, modifying them, verifying constraints), and TravelPlanner requires multi-step planning with complex constraint satisfaction.
>
> In addition to these two in-domain benchmarks, we evaluate on **8 out-of-domain benchmarks** (7 GEM datasets + τ-Bench Airline), demonstrating broad generalization.
>
> SWE-Bench and WebArena involve significantly higher computational costs per evaluation (full Docker environments, hour-long trajectories), which would make our extensive ablation study (3 PI types × 3 α values × 4+ β values × 3 models × 3 seeds) prohibitively expensive. We believe our current evaluation breadth (2 training domains, 8 OOD benchmarks, 3 model families) provides strong evidence for π-Distill's generality, and we hope future work extends to other settings.
>
> ---
>
> ### W3: Hyperparameter tuning (α and β)
>
> While α is an important hyperparameter, our analysis in Section 7 and Figure 2 characterizes precisely when different values of α succeed and fail. Crucially, we show that **α = 0.5 is the most robust configuration**: it achieves the best performance in 6 out of 16 scenarios and effectively avoids the failure modes of the other variants, never ranking as the worst performer (Section 7). Thus, while tuning α can yield marginal gains, α = 0.5 provides a strong, reliable default that substantially outperforms all baselines.
>
> Regarding β, this hyperparameter serves the same role as the KL penalty coefficient in standard KL-regularized RL (e.g., RLHF). We show in Section 8 and Figure 8 that **β > 0 achieves top performance in 8/9 settings**, meaning the practitioner simply needs to set β > 0 and the exact value matters less. Thus, π-Distill does not introduce fundamentally new tuning complexity beyond what is already standard in post-training pipelines.
>
> ---
>
> ### Q1: Equation typo
>
> Thank you for catching this! We have fixed this and it will be reflected in our next revision.
>
> ---
>
> ### Q2: Importance sampling validity with narrower teacher distribution
>
>
> The reviewer raises an important theoretical concern. We address this in three ways:
>
> **First**, empirically, we find that the entropy of the teacher and student policies are fairly similar in practice, mitigating the severity of this issue.
>
> **Second**, this concern is precisely one of the motivations for our study of three different PI types with varying degrees of distributional shift (Section 7). We find that higher KL divergence between teacher and student (induced by more informative but more off-policy PI) does degrade performance when using student-only training (α = 0), consistent with the reviewer's intuition.
>
> **Third**, this is where π-Distill's joint training (α = 0.5) and teacher-only training (α = 1) provide a solution: by also optimizing the teacher objective, the teacher is encouraged to stay close to the student policy (via the reverse KL penalty), which directly mitigates the distribution mismatch. As shown in Figure 2, joint training (α = 0.5) is the most robust to high KL settings, and teacher training (α = 1) can recover even when the initial PI utility (Δ) is negative.
>
> We remain open and happy to clarify any further concerns.

---

> > ### Author Rebuttal · Reviewer_ELnF · 2026-04-04
> >
> > I think my concern about evaluation on the multi-turn dataset still remains. So I decide to keep my score unchanged.

---

> > > ### Author Response · Authors · 2026-04-05
> > >
> > > Dear Reviewer,
> > >
> > > We want to clarify a potential misunderstanding: τ-Bench [1] and TravelPlanner [2] are both multi-turn datasets, with tasks spanning 10–20 turns on average. Combined with τ-Bench Airline, we evaluate on **3 multi-turn benchmarks** in total, plus 8 out-of-domain benchmarks.
> > >
> > > We believe this directly addresses the core concern, and kindly ask the reviewer to reconsider their score.
> > >
> > > [1] τ-bench: A Benchmark for Tool-Agent-User Interaction in Real-World Domains (https://arxiv.org/abs/2406.12045)
> > >
> > >
> > > [2] TravelPlanner: A Benchmark for Real-World Planning with Language Agents (https://arxiv.org/abs/2402.01622)

---

### Decision · Program_Chairs · 2026-04-30

**Decision:**

Accept (regular)

**Comment:**

This paper introduces \pi-Distill, a novel and highly relevant joint teacher-student reinforcement learning framework that distills agentic capabilities from frontier language models using Privileged Information (PI), effectively eliminating the need for full Chain-of-Thought (CoT) traces. The reviewers acknowledge the method's theoretical soundness and strong empirical results, highlighting that \pi-Distill consistently matches or outperforms standard SFT and RL baselines—even those acting as soft upper bounds with full CoT access—across complex, multi-turn domains like \tau-Bench and TravelPlanner, as well as on several out-of-distribution (OOD) tasks.

While initial reviews raised valid questions regarding structural organization, hyper-parameter sensitivity (specifically regarding \alpha and \beta), computational overhead, and comparisons to concurrent self-distillation literature, the authors provided a comprehensive and compelling rebuttal. They successfully demonstrated the robustness of the joint training approach in high-KL regimes, clarified the remarkable transfer efficiency relative to the teacher model, and updated their baselines to distinguish their framework from simpler on-policy context distillation methods. By presenting a rigorously ablated and highly practical solution to the black-box distillation problem, this work offers a substantial contribution, justifying its acceptance.